# Avoid Catastrophic Forgetting with Rank-1 Fisher from Diffusion Models

**Zekun Wang**[*]  **Anant Gupta**[*]  **Zihan Dong**  **Christopher J. MacLellan**
College of Computing
Georgia Institute of Technology
Atlanta, GA 30332, USA
`{zekun,agupta886,zdong312,cmaclell}@gatech.edu`

## Abstract

Catastrophic forgetting remains a central obstacle for continual learning in neural models. Popular approaches—replay and elastic weight consolidation (EWC)—have limitations: replay requires a strong generator and is prone to distributional drift, while EWC implicitly assumes a shared optimum across tasks and typically uses a diagonal Fisher approximation. In this work, we study the gradient geometry of diffusion models, which can already produce high-quality replay data. We provide theoretical and empirical evidence that, in the low signal-to-noise ratio (SNR) regime, per-sample gradients become strongly collinear, yielding an empirical Fisher that is effectively rank-1 and aligned with the mean gradient. Leveraging this structure, we propose a rank-1 variant of EWC that is as cheap as the diagonal approximation yet captures the dominant curvature direction. We pair this penalty with a replay-based approach to encourage parameter sharing across tasks while mitigating drift. On class-incremental image generation datasets (MNIST, FashionMNIST, CIFAR-10, ImageNet-1k), our method consistently improves average FID and reduces forgetting relative to replay-only and diagonal-EWC baselines. In particular, forgetting is nearly eliminated on MNIST and FashionMNIST and is more than halved on ImageNet-1k. These results suggest that diffusion models admit an approximately rank-1 Fisher. With a better Fisher estimate, EWC becomes a strong complement to replay: replay encourages parameter sharing across tasks, while EWC effectively constrains replay-induced drift.

## 1 Introduction

The task of continual learning aims to train neural models on a stream of tasks without revisiting the full past data. A long–standing obstacle is catastrophic forgetting—when learning new tasks drastically degrades performance on earlier ones (McCloskey & Cohen, 1989; French, 1999; Parisi et al., 2019). In contrast, humans exhibit a striking robustness to interference, partly due to systems-level mechanisms such as memory consolidation and replay in the hippocampal–neocortical loop (McClelland et al., 1995; McGaugh, 2000; Foster & Wilson, 2006). These observations have inspired a family of continual learning methods that explicitly encode consolidation or replay in training deep networks on continuous tasks.

Two representative approaches are elastic weight consolidation (EWC) (Kirkpatrick et al., 2017) and generative replay (Shin et al., 2017). EWC constrains parameter changes with a quadratic penalty weighted by an estimate of the Fisher information at a previously learned task, constraining updates to remain near parameter directions that support old tasks (Kirkpatrick et al., 2017; Schwarz et al., 2018). Intuitively, EWC behaves like an online consolidation process that selectively "stiffens" important parameters. Generative replay maintains a generator and distills samples from past tasks while learning new ones, thereby approximating rehearsal without storing original data (Shin et al., 2017; van de Ven & Tolias, 2018). However, replay inherits the generator's imperfections and can amplify distributional shift. EWC in practice relies on a diagonal Fisher approximation, which

---

[*]Equal contribution.
[†]Code is available at: `https://github.com/Teachable-AI-Lab/iclr2026-rank1-fisher`

neglects cross-parameter correlations and struggles to find a shared parameter space between tasks in overparameterized models, particularly when tasks have disjoint optimum. These observations suggest a complementary pairing: replay can encourage parameter sharing by exposing shared data support, while a stronger approximation to Fisher information can constrain updates and mitigate replay's residual shift.

In this work, we study the gradient behavior of diffusion models (Ho et al., 2020; Nichol & Dhariwal, 2021), which are already capable of generating high quality replay data. Our starting point is the observation that diffusion models admit a tractable gradient structure when the signal-to-noise ratio (SNR) is lower (at later timesteps). As a model converges, the per-sample gradients $g$ become approximately collinear with their mean $u$. This makes the empirical Fisher $F$ effectively rank-1:

$$F \;=\; \mathbb{E}\big[g\,g^\top\big] \;\approx\; \alpha\,u\,u^\top, \quad u = \mathbb{E}[g],$$

This yields a consolidation penalty that captures the dominant curvature direction "for free" from model gradients. In contrast, the commonly used diagonal Fisher approximation captures almost no curvature when SNR is low. We leverage this structure to instantiate a rank-1 EWC that complements generative replay: replay encourages cross-task parameter overlap while our proposed rank-1 EWC constrains updates along the principal sensitive direction to the shared optimum across tasks.

Our main contributions are: (1) We provide both theoretical and empirical characterizations of Fisher information geometry in diffusion models, showing that low SNR induces a near rank-1 Fisher aligned with the mean gradient. (2) We propose a practical rank-1 EWC penalty that is as cheap as a diagonal penalty but captures more curvature information for diffusion models. (3) We demonstrate that combining rank-1 EWC with distillation-based replay substantially reduces forgetting in continual image generation tasks, improving generation fidelity and stability across long horizons. On MNIST, FashionMNIST, CIFAR-10, and ImageNet-1k, our method consistently outperforms replay-only and diagonal Fisher baselines. In particular, forgetting is nearly eliminated on MNIST and FashionMNIST and is more than halved on ImageNet-1k, the longest-horizon setting, relative to baselines.

## 2 BACKGROUND AND RELATED WORK

### 2.1 DIFFUSION MODELS

Diffusion models are a family of generative models that define a fixed forward noising process and learn a reverse (denoising) process that maps Gaussian noise to data (Sohl-Dickstein et al., 2015; Ho et al., 2020). The forward chain corrupts data $x_0 \sim q_0$ via a Markov process

$$q(x_t \mid x_{t-1}) \;\sim\; \mathcal{N}\big(\sqrt{1-\beta_t}\,x_{t-1},\, \beta_t \mathbf{I}\big), \quad q(x_t \mid x_0) \;\sim\; \mathcal{N}\big(\sqrt{\bar{\alpha}_t}\,x_0,\, (1-\bar{\alpha}_t)\mathbf{I}\big),$$

where $\alpha_t = 1 - \beta_t$, $\bar{\alpha}_t = \prod_{s=1}^{t} \alpha_s$, $\beta_t \in (0, 1)$ is the noise level, and $t \in \{0 \cdots N\}$ is the discrete forward noising process timestep. The reverse transitions $p_\theta(x_{t-1} \mid x_t)$ are parameterized by a neural network $\varepsilon_\theta$ and trained by maximizing a variational lower bound (ELBO). In practice, the ELBO reduces to a reweighted denoising loss (Ho et al., 2020):

$$\mathcal{L}_{\text{simple}}(\theta) \;=\; \frac{1}{2}\mathbb{E}_{t,\, x_0,\, \varepsilon \sim \mathcal{N}(0,\mathbf{I})}\Big[\, \big\|\varepsilon - \varepsilon_\theta(x_t, t)\big\|_2^2 \,\Big], \quad x_t = \sqrt{\bar{\alpha}_t}x_0 + \sqrt{1-\bar{\alpha}_t}\,\varepsilon.$$

This surrogate loss can be interpreted as a score-based generative modeling that estimates the score $\nabla_{x_t} \log q_t(x_t)$ at each timestep, where $q_t(x_t) = \int q(x_t \mid x_0)q_0(x_0)dx_0$, and then integrates a reverse-time SDE/ODE to sample (Song & Ermon, 2019; Song et al., 2021b; Vincent, 2011; Hyvärinen, 2005): $\nabla_{x_t} \log q_t(x_t) = -\frac{1}{\sqrt{1-\bar{\alpha}_t}} \mathbb{E}[\varepsilon \mid x_t]$, and the model's score estimate follows from $s_\theta(x_t, t) \;=\; -\frac{1}{\sqrt{1-\bar{\alpha}_t}} \varepsilon_\theta(x_t, t)$, where $\varepsilon_\theta(x_t, t) = \mathbb{E}[\varepsilon \mid x_t]$ at model optimum.

Denoising Diffusion Implicit Models (Song et al., 2021a) further show that one can define a non-Markovian, deterministic sampling path that preserves DDPM's per-time marginal distributions while allowing far fewer steps. In this work, we use a DDIM sampling process for faster image generation.

### 2.2 ELASTIC WEIGHT CONSOLIDATION

Elastic weight consolidation (Kirkpatrick et al., 2017) mitigates catastrophic forgetting by regularizing parameter updates using information about how important each weight was to previously learned

tasks. It casts continual learning as approximate Bayesian updating: for tasks $1{:}T$, the posterior factors as $p(\theta \mid \mathcal{D}_{1:T}) \propto p(\mathcal{D}_T \mid \theta)\, p(\theta \mid \mathcal{D}_{1:T-1})$. EWC approximates the previous-task posterior $p(\theta \mid \mathcal{D}_{1:T-1})$ with a Laplace approximation around the prior optimum $\theta_{T-1}^{\star}$, yielding $-\log p(\theta \mid \mathcal{D}_{1:T-1}) \approx \frac{1}{2}(\theta - \theta_{T-1}^{\star})^{\top} F^{(T-1)}(\theta - \theta_{T-1}^{\star})$, where $F^{(T-1)}$ is the Fisher information evaluated at $\theta_{T-1}^{\star}$. Plugging this quadratic surrogate into the negative log-posterior for task $k$ gives the EWC objective

$$\mathcal{L}_{\mathrm{EWC}}(\theta) \;=\; \mathcal{L}_T(\theta) \;+\; \frac{\lambda}{2} \sum_{k=1}^{T-1} (\theta - \theta_k^{\star})^{\top} F^{(k)} (\theta - \theta_k^{\star}), \tag{1}$$

with $\lambda$ trading off plasticity and stability. Intuitively, parameters with high curvature under the previous task are penalized more strongly, discouraging changes that would degrade old performance. In the original work, $\mathcal{L}_T(\theta)$ is the negative log-likelihood for a classification model. For latent-variable generative models such as diffusion models, one can replace this term with a tractable variational surrogate such as the negative ELBO.

In practice, a diagonal Fisher is often used as an approximation to the full Fisher. In this work, we show that diffusion models approximate a rank-1 Fisher for free for a more effective application in continual learning.

## 2.3 CONTINUAL LEARNING WITH GENERATIVE MODELS

Continual learning trains a model on a sequence of tasks, posing the dual challenge of adapting to distribution shift while preserving knowledge from earlier tasks. Broadly, continual learning approaches fall into (i) regularization-based constraints on parameter drift (e.g., EWC) and (ii) replay-based strategies that rehearse past knowledge (Kirkpatrick et al., 2017; van de Ven & Tolias, 2018). For generative models, especially diffusion models, replay is particularly natural because the model can synthesize high-fidelity samples for rehearsal (Ho et al., 2020).

DDGR uses a diffusion generator with class conditioning to synthesize exemplars of prior tasks, and diffusion-based replay has been adapted to dense prediction (segmentation, detection) using task-specific guidance or pseudo-labels (Gao et al., 2023; Chen et al., 2023; Kim et al., 2024). However, naïve replay with continually updated diffusion models can degrade denoising because the reverse process itself drifts across tasks. Recent work on generative distillation mitigates this by distilling the entire reverse chain ($\theta_{t-1}^* \to \theta_t$) across timesteps, aligning noise predictions so the continually trained model retains both sample quality and coverage of past tasks (Masip et al., 2025). In this work, we employ generative distillation along with our approach to EWC for a more effective learning.

## 3 DIFFUSION MODELS APPROXIMATE THE RANK-1 FISHER

The Bayesian view of the continual learning suggests that $p(\theta \mid \mathcal{D}_{1:T}) \propto p(\mathcal{D}_T \mid \theta)\, p(\theta \mid \mathcal{D}_{1:T-1})$ where $\mathcal{D}_T$ is the dataset for task $T$ and $\theta$ represents the model parameters. EWC (Kirkpatrick et al., 2017) proposed a Laplace approximation of the posterior distribution $p(\theta \mid \mathcal{D}_{1:T-1})$ for previous tasks as $-\log p(\theta \mid \mathcal{D}_{1:T-1}) \approx \frac{1}{2}(\theta - \theta_{T-1}^{\star})^{\top} F^{(T-1)}(\theta - \theta_{T-1}^{\star})$, where $F^{(T-1)}$ is the Fisher information matrix for the previous task. However, forming a full Fisher is impractical and a diagonal approximation is widely used. In this work, we show that the Fisher of a diffusion model can be approximated as a rank-1 structure, which better captures important weights associated with previous tasks. We theoretically analyze why diffusion models approximate a rank-1 Fisher in Section 3.1 along with empirical analysis in Section 3.2. We then derive the practical EWC loss using rank-1 Fisher in Section 3.3 and complement it with generative distillation in Section 3.4.

### 3.1 PER-SAMPLE GRADIENTS ALIGN WITH THEIR MEAN

**Setup and notations.** We use the standard variance-preserving forward process

$$x_t \;=\; \sqrt{\bar{\alpha}_t}\, x_0 \;+\; \sqrt{1 - \bar{\alpha}_t}\, \varepsilon, \quad \varepsilon \sim \mathcal{N}(0, I), \quad t \in \{1 \cdots N\}$$

If we let $\varepsilon_\theta(x_t, t)$ be the noise prediction network, the per-sample surrogate loss (Ho et al., 2020) is:

$$\mathcal{L}_{\mathrm{simple}}(\theta; x_t) \;=\; \tfrac{1}{2} \left\| \varepsilon - \varepsilon_\theta(x_t, t) \right\|_2^2. \tag{2}$$

From a denoise score-matching perspective, let model score $s_\theta(x_t, t) := \nabla_{x_t} \log p_\theta(x_t)$ and the true marginal score $s_t^\star(x_t) := \nabla_{x_t} \log q_t(x_t)$, where $q_t$ denotes the noisy data distribution at time $t$ and let $p_\theta$ be the model with parameters $\theta$. The connection to per-sample $\mathcal{L}_{\text{simple}}$ can expressed as follows (Ho et al., 2020; Vincent, 2011; Song et al., 2021b):

$$\varepsilon_\theta(x_t, t) = -\sqrt{1 - \bar{\alpha}_t} \, s_\theta(x_t, t), \quad \mathbb{E}[\varepsilon \mid x_t] = -\sqrt{1 - \bar{\alpha}_t} \, s_t^\star(x_t) \tag{3}$$

Substituting Equation 3 into Equation 2 yields the per-sample denoising score-matching loss:

$$\mathcal{L}_{DSM}(\theta; x_t) = \frac{1 - \bar{\alpha}_t}{2} \left\| s_\theta(x_t, t) - s_t^\star(x_t) \right\|_2^2, \tag{4}$$

Additionally, we define $\sqrt{\bar{\alpha}_t}$ as the signal level and $1 - \bar{\alpha}_t$ as the noise level at time $t$. The signal-to-noise ratio $SNR := \sqrt{\bar{\alpha}_t}/(1 - \bar{\alpha}_t)$ decreases with $t$. Intuitively, noise will dominate the signal in the later diffusion timesteps.

**Proposition 1.** *Let $s_t^\star(x_t)$, $x_t \sim q_t$, be the score of the noisy data distribution at time $t$ in a variance-preserving diffusion process. As SNR decreases, $s_t^\star(x_t) \approx -x_t/(1 - \bar{\alpha}_t)$.*

We refer Appendix B.1 for the proof. Proposition 1 shows that in the low SNR region (i.e, at later diffusion timesteps), the score function is approximately predicting its scaled input. In other words, the score function behaves like a scaled identity map from $x_t$ to $-x_t/(1 - \bar{\alpha}_t)$ when the SNR is low.

**Assumption 1.** $s_\theta(x_t, t)$ *approximates a linear function* $s_\theta(x_t, t) \approx A_\theta x_t$, *where $A_\theta$ is a linear operator, when the model learns to perform scaled identity mapping such that $s_\theta(x_t, t) = x_t \gamma_t$, $\gamma_t \in \mathbb{R}$.*

**Proposition 2.** *As the SNR decreases and the model converges, for any $x_t \sim q_t$, the per-sample gradient $\nabla_\theta \mathcal{L}_{\text{DSM}}(\theta; x_t)$ becomes collinear with its population mean under Assumption 1:*

$$\nabla_\theta \mathcal{L}_{\text{DSM}}(\theta; x_t) \propto \mathbb{E}_{x_t' \sim q_t}[\nabla_\theta \mathcal{L}_{\text{DSM}}(\theta; x_t')].$$

Proposition 1 suggests that when the SNR is low, $s_t^\star(x_t) \approx x_t \gamma_t$, where $\gamma_t = -1/(1 - \bar{\alpha}_t)$, which is a scalar independent of input $x_t$. Near convergence, $s_\theta(x_t, t) \approx s_t^\star(x_t) \approx x_t \gamma_t$. Assumption 1 hypothesizes that $s_\theta(x_t, t)$ could be linear and takes the form of $s_\theta(x_t, t) = A_\theta x_t$. This is plausible for an autoencoder with nonlinearities optimized with the MSE objective, such as the $\mathcal{L}_{\text{simple}}$ in Ho et al. (2020), as these autoencoders are still "likely" to lie in a PCA linear subspace (Vincent et al., 2010). We leave additional discussion and analysis on alternative autoencoding objectives in Appendix C. At a given $t$ such that the SNR is low, $s_\theta(x_t, t) \approx A_\theta x_t \approx x_t \gamma_t$. As a result, the gradient of $s_\theta(x_t, t)$ can be understood as the direction that moves $A_\theta$ to $\gamma_t I$. Since $s_\theta(x_t, t) \approx A_\theta x_t$ is linear, the directional change in $A_\theta$ at each $x_t$ is collinear to $A_\theta - \gamma_t I$ independent of $x_t$, so the per-sample gradients are collinear with each other, and hence collinear to their mean. We leave the complete proof in Appendix B.2.

**Theorem 1.** *Under Proposition 1 and Proposition 2, the empirical Fisher information matrix at time $t$: $F_t(\theta) = \mathbb{E}_{x_t' \sim q_t}[g(x_t'; \theta) \, g(x_t'; \theta)^\top]$, $g(x_t; \theta) = \nabla_\theta \mathcal{L}_{DSM}(\theta; x_t)$, is approximately rank-1 when the SNR is low with eigenvector $\mu_t(\theta) = \mathbb{E}_{x_t' \sim q_t}[g(x_t'; \theta)]$, and eigenvalue $\frac{\mu_t(\theta)^\top F_t(\theta) \mu_t(\theta)}{\|\mu_t(\theta)\|_2^4}$.*

*Proof.* Let $v_t(\theta) = \mathbb{E}_{x_t' \sim q_t}[g(x_t'; \theta)]$ be the mean of the gradients. By Proposition 2, $g(x_t; \theta) \approx c(x_t) v_t(\theta)$ for some scalar function $c(\cdot)$. Then

$$F_t(\theta) = \mathbb{E}_{x_t' \sim q_t}[g(x_t'; \theta) \, g(x_t'; \theta)^\top] \approx \mathbb{E}_{x_t' \sim q_t}[c^2(x_t')] v_t(\theta) v_t(\theta)^\top \tag{5}$$

Therefore, $F_t(\theta)$ is approximately rank-1 with eigenvector $v_t(\theta) = \mu_t(\theta)$ and eigenvalue $\mathbb{E}_{x_t' \sim q_t}[c^2(x_t')]$. Multiplying 5 by $\mu^\top$ on the left and $\mu$ on the right, we get

$$\mu_t^\top(\theta) F_t(\theta) \mu_t(\theta) \approx \mathbb{E}_{x_t' \sim q_t}[c^2(x_t')] \|\mu_t(\theta)\|^4 \implies \mathbb{E}_{x_t' \sim q_t}[c^2(x_t')] \approx \frac{\mu_t(\theta)^\top F_t(\theta) \mu_t(\theta)}{\|\mu_t(\theta)\|^4}$$

$\square$

## 3.2 EMPIRICAL VALIDATION

Theorem 1 suggests that, under model convergence, diffusion models approximate a rank-1 Fisher information matrix for their gradients as the SNR decreases (i.e., as the forward process timestep increases). To empirically verify this result, we train a small diffusion model on MNIST (Lecun et al., 1998), whose full Fisher matrix fits in GPU memory, and analyze its gradient behavior across timesteps $t = 100, 200, \ldots, 900$. For each $t$, we sample 1024 data points and compute their gradients. Experiment and model details can be found in Appendix D.

We first empirically validate Proposition 1, which states that when SNR is low, the denoising network behaves as a scaled identity map. Let $\varepsilon_\theta(x_t; t) = \hat{\varepsilon}$ be the denoising network, and by Tweedie's identity (Efron, 2011), $\hat{x}_t = \sqrt{1 - \bar{\alpha}_t}\hat{\varepsilon}$. Figure 1 plots the mean-square error between input $x_t$ and prediction $\hat{x}_t$. As timestep increases (SNR decreases), the model begins to perfectly predict its scaled input with error approaching 0.

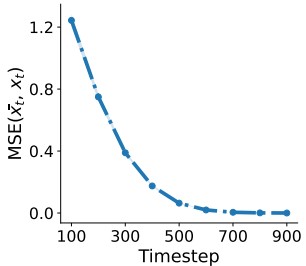

Figure 1: MSE between model input $x_t$ and the scaled prediction $\hat{x}_t$ at each timestep.

We then empirically validate Proposition 2 that the per-sample gradients $g(\theta; x_t)$ are collinear with their mean $\mu_t(\theta) = \mathbb{E}_{x'_t \sim q_t}[g(\theta; x'_t)]$ at low SNR when the model converges. For each gradient, we compute the absolute cosine similarity to the population mean. Figure 2 shows these similarities at selected timesteps where each pixel represents a per-sample similarity. We find that per-sample gradients are mostly collinear with their mean at each timestep, with stronger collinearity in the mid-to-late timesteps (i.e. deeper red). This also provides empirical support for Assumption 1 that $s_\theta(x_t, t) \approx x_t\theta$ such that the per-sample gradient becomes a scaled directional change in $\Delta\theta = \theta - \gamma_t I$ that is not dependent on $x_t$. We additionally plot the pairwise cosine similarities of $\mu_t(\theta)$ across timesteps in Figure 3a. We find that the $\mu_t(\theta)$ for different timesteps highly align with each other. This suggests a practical benefit where we can Monte-Carlo sample timesteps instead of constructing a separate Fisher $F_t(\theta)$ at each timestep.

To probe the rank of the Fisher, we perform eigen decomposition on the empirical Fisher $F_t(\theta)$ from the gradients of the 1024 samples and collect top 5 eigenvalues $\lambda_1 \geq \cdots \geq \lambda_5$. If $F_t(\theta)$ is nearly

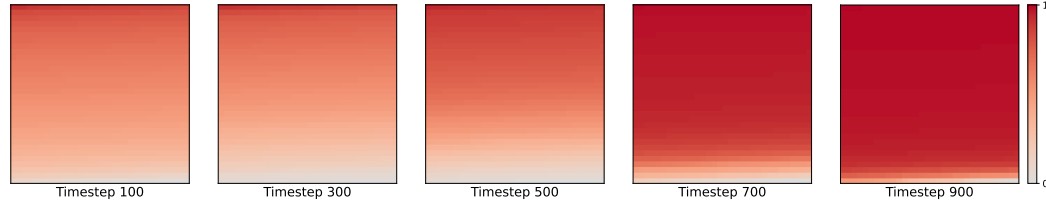

Figure 2: Absolute cosine similarities between per-sample gradient $g(\theta; x_t)$ and their expectation $\mu(\theta)$ at different diffusion timesteps. Each pixel represents a per-sample similarity. Higher values (deeper red) indicate stronger collinearity with $\mu(\theta)$.

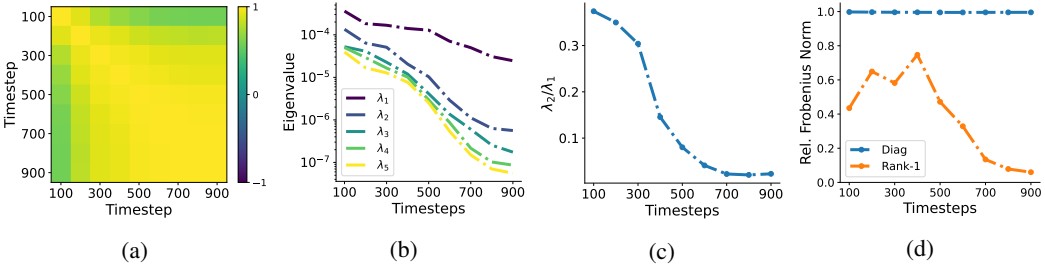

Figure 3: (a): Pairwise cosine similarities of $\mu_t(\theta)$ across each forward process timestep. (b): Top 5 eigenvalues of $F_t(\theta)$ across timesteps in log-scale. (c): The ratio $r_t = \lambda_2/\lambda_1$ across timesteps. (d): Relative Frobenius norm between $F_t(\theta)$ and diagonal and rank-1 approximations across timesteps.

rank-1, then $\lambda_1 \gg \lambda_2$. In Figure 3b, we plot the top 5 eigenvalues at each timestep in log-scale. We observe that $\lambda_1$ is typically one or two orders of magnitude larger than the remaining eigenvalues and their overall magnitude decreases as timestep increases. To quantify the dominance of the leading eigenvalue, we compute the ratio $r_t = \lambda_2/\lambda_1$. Smaller $r_t$ indicates a larger eigengap and stronger rank-1 behavior. Figure 3c indicates that $\lambda_1 \gg \lambda_2$ at all timesteps with the lowest ratio ($r_t = 0.022$) achieved at $t = 700$, suggesting a sharper single-eigenvalue dominance in the low SNR timesteps.

Since Fisher is empirically near rank-1, we compare two approximations at each timestep: (i) the rank-1 reconstruction from Equation 5, and (ii) the diagonal $F_t^{\text{diag}}(\theta)$. For each, we report the relative Frobenius error with respect to the full Fisher: $\text{err}(\widehat{F}_t) = \frac{\|F_t(\theta) - \widehat{F}_t(\theta)\|_F}{\|F_t(\theta)\|_F}$, $\widehat{F}_t \in \{F_t^{\text{rank1}}, F_t^{\text{diag}}\}$, where $\|\cdot\|_F$ is the Frobenius norm. Figure 3d shows that rank-1 approximation achieves a lower error at the mid-to-late timesteps, suggesting that the rank-1 reconstruction is close to the true Fisher. This result aligns with the above two empirical results that the rank-1 structure is clearer in the low SNR timesteps. It is worth noting that the diagonal approximation yields an error near $1.0$ at every timestep. This behavior indicates that the Fisher information matrix in diffusion models has most of its curvature concentrated in the off-diagonal terms, making the diagonal approximation particularly inadequate.

Our empirical results collectively suggest that the per-sample gradients are mostly collinear with their mean as SNR decreases and hence approximates a rank-1 structure that captures most of the curvatures in Fisher where a diagonal approximation fails.

### 3.3 EWC PENALTY WITH RANK-1 FISHER

With the rank-1 approximation using Equation 5, we derive a practical EWC penalty term without forming a full Fisher matrix at each timestep and only use the gradients from the model. For simplicity of notation, let $g = g(\theta; x_t)$ and $\mu = \mu(\theta) = \mathbb{E}[g]$. In implementation, this expectation is taken over the joint sampling process used during diffusion training: we draw a datapoint $x_0 \sim p_{\text{data}}$, sample a timestep $t \sim p(t)$ (uniform in our experiments unless stated otherwise), construct the noised input $x_t$ from $(x_0, t)$, and then compute $g(\theta; x_t, t) = \nabla_\theta \ell(\theta; x_t, t)$. Thus, $\mu$ should be understood as $\mu(\theta) = \mathbb{E}_{x_0, t}[g(\theta; x_t, t)]$

This comes from the observation that the *SNR* decreases sharply at early timesteps via Tweedie's identity, and therefore remains low for the majority of diffusion steps (see Figure 1). Consistent with this, the empirical Fisher is strongly dominated by its leading eigen-direction across timesteps. For instance, we observe $\lambda_2/\lambda_1 < 0.4$ already at $t = 100$ in Figure 3c, and the mean gradient at different timesteps is highly collinear with one another in Figure 3a. This indicates that the rank-1 approximation captures a large portion of the curvature even outside the latest timesteps. Consequently, averaging gradients over timesteps yields a practical surrogate that dilutes potentially residual higher-rank structure from the earliest steps and is dominated by the nearly rank-1 gradients from the remaining timesteps. With $F(\theta) = \mathbb{E}[gg^\top]$, we have

$$\mu^\top F(\theta)\mu = \mu^\top \mathbb{E}[gg^\top]\mu = \mathbb{E}[(\mu^\top g)^2] \quad \Rightarrow \quad c^\star = \mathbb{E}[c_\theta^2(x_t)] = \frac{\mu^\top F(\theta)\mu}{\|\mu\|^4} = \frac{\mathbb{E}[(\mu^\top g)^2]}{\|\mu\|^4}.$$

Thus, plugging in $c^\star$ and Equation 5 into Equation 1, the EWC penalty with the rank-1 Fisher is:

$$\mathcal{L}_{\text{Rank-1}}(\theta) = \mathcal{L}_T(\theta) + \frac{\lambda}{2} \sum_{k=1}^{T-1} c_k^\star \left(\mu_k^\top (\theta - \theta_k^\star)\right)^2. \tag{6}$$

### 3.4 PROMOTING PARAMETER SHARING ACROSS TASKS VIA GENERATIVE DISTILLATION

EWC is most effective when the task optima lie in a shared parameter subspace, so that its quadratic penalty can steer gradient descent to a region that remains good for all tasks. In overparameterized models, however, different tasks can converge to disjoint basins, in which case no single parameter vector is simultaneously optimal, and the EWC penalty alone may fail.

To mitigate this limitation and strengthen our evaluation, we promote parameter sharing by adding a generative distillation term computed on replayed inputs from earlier tasks (Masip et al., 2025). Concretely, we keep a frozen teacher model $\varepsilon_{\theta_{T-1}^\star}$ from the previous task and sample replay inputs $\tilde{x}$

from it. Generative distillation then encourages the current model $\varepsilon_\theta$ to match the teacher model's denoising behavior:

$$\mathcal{L}_{\text{GD}}(\theta) \; = \; \mathbb{E}_{\tilde{x} \sim \tilde{\mathcal{D}}} \left[ \frac{1}{2} \big\| \varepsilon_\theta(\tilde{x}) - \varepsilon_{\theta^\star_{T-1}}(\tilde{x}) \big\|_2^2 \right],$$

where $\tilde{\mathcal{D}}$ denotes the replay distribution. Our full objective becomes:

$$\mathcal{L}_{\text{total}}(\theta) = \mathcal{L}_{\text{Rank-1}}(\theta) + \mathcal{L}_{\text{GD}}(\theta) \tag{7}$$

Intuitively, the generative distillation term pulls $\varepsilon_\theta$ to remain compatible with past task behaviors on their input manifolds, guiding gradient descent toward regions that overlap with previous optima, thereby complementing EWC's curvature-based constraint.

## 4    CONTINUAL LEARNING WITH RANK-1 FISHER

As the Fisher of diffusion models is approximately rank-1, we validate the effectiveness of our findings on class-incremental continual learning tasks.

### 4.1    DATASETS FOR CLASS-INCREMENTAL CONTINUAL LEARNING

In class-incremental continual learning, a dataset is partitioned into $T$ tasks, where each task $T_k$ contains $n$ class labels and $\cap T_k = \emptyset$. We evaluate our approach on four image datasets commonly used in generative modeling: MNIST (Lecun et al., 1998), Fashion MNIST (FMNIST) (Xiao et al., 2017), CIFAR-10 (Krizhevsky, 2009), and ImageNet-1k (Deng et al., 2009). We use the down-sampled ImageNet-1k (Chrabaszcz et al., 2017) such that each image has the dimension of $3 \times 32 \times 32$ for faster training and evaluation while preserving similar performance characteristics to the full-size ImageNet-1k. We additionally pad MNIST and FMNIST to $32 \times 32$ to be consistent with CIFAR-10 and down-sampled ImageNet-1k. For each of MNIST, FMNIST, and CIFAR-10, we partition the dataset into 5 tasks with 2 classes per task. For ImageNet-1k, we partition it into 20 tasks with 50 classes per task, simulating a much longer horizon continual learning task. We use the same class label ordering as in the original dataset. We refer to Appendix E.3 for additional details.

### 4.2    BASELINES AND METRICS

We compare our proposed EWC approach with a rank-1 Fisher approximation (Rank-1) against the widely used diagonal Fisher approximation (Diag), with both methods augmented by generative distillation as suggested in Section 3.4. For the ablation study, we compare both EWC approaches without generative distillation. In addition, we evaluate the generative distillation only (GD) approach to assess whether EWC provides complementary benefits beyond distillation. Finally, we compare these approaches to the non-continual learning setting, which serves as an upper bound for the performance. We hypothesize that with parameter sharing across tasks encouraged by generative distillation, a better Fisher approximation approach will lead to better continual learning performance and will complement generative distillation.

**Metrics.** To evaluate continual learning performance for generative models, we compute the Fréchet Inception Distance (FID; Heusel et al., 2018) between generated samples and each task's held-out test set. Let $m_k$ be the model after training on tasks $1{:}k$, and let $\text{FID}_i(m_k)$ denote the FID on task $i$'s test set when evaluated with $m_k$. We report two primary metrics:

- **Average FID through task $k$.** $\mathcal{A}\text{FID@}k \;=\; \frac{1}{k} \sum_{i=1}^k \text{FID}_i(m_k)$, so $\mathcal{A}\text{FID@}T$ summarizes overall performance at the end of training.
- **Final average forgetting.** $\mathcal{F} \;=\; \frac{1}{T} \sum_{k=1}^T \Big( \text{FID}_k(m_T) - \text{FID}_k(m_k) \Big)$, the mean change in each task's FID immediately after it is learned ($m_k$) as compared to after training on all tasks ($m_T$).

### 4.3    IMPLEMENTATION DETAILS

We use the label conditioning UNet implementation from *Huggingface* with default hyper-parameters as our denoising network. We use 4 ResNet blocks with 128 output channels in the first down-sample block and 256 output channels in the rest. For sampling, we use a DDIM scheduler with 50 sampling

steps and 1000 noising steps. We train our UNet with 100 epochs on ImageNet-1k and 200 epochs on the rest of the datasets for each task with a batch size of 128 and a learning rate of $2 \times 10^{-4}$ using Adam (Kingma & Ba, 2017). We use an EWC penalty weight of 15000 for all tasks. For generative distillation, we create a replay buffer of 1300 images per class for ImageNet-1k and 5000 images per class for other datasets. We refer to Appendix D for additional training details.

# 5 RESULTS AND DISCUSSIONS

## 5.1 CONTINUAL LEARNING PERFORMANCE

Table 1: Average FID at the final task and average forgetting across methods and datasets. Standard errors are reported over 3 random seeds.

| Methods | MNIST | | FMNIST | | CIFAR-10 | | ImageNet-1k | |
|---|---|---|---|---|---|---|---|---|
| | $\mathcal{A}$FID$\downarrow$ | $\mathcal{F}\downarrow$ | $\mathcal{A}$FID$\downarrow$ | $\mathcal{F}\downarrow$ | $\mathcal{A}$FID$\downarrow$ | $\mathcal{F}\downarrow$ | $\mathcal{A}$FID$\downarrow$ | $\mathcal{F}\downarrow$ |
| Non-continual | $2.6_{\pm0.1}$ | – | $5.7_{\pm0.8}$ | – | $23.3_{\pm0.7}$ | – | $11.7_{\pm0.1}$ | – |
| Diag$_{\text{w/o GD}}$ | $62.2_{\pm2.9}$ | $51.1_{\pm4.2}$ | $99.9_{\pm3.5}$ | $81.7_{\pm4.7}$ | $128.6_{\pm4.6}$ | $74.4_{\pm3.5}$ | $86.1_{\pm4.2}$ | $34.2_{\pm3.6}$ |
| Rank-1$_{\text{w/o GD}}$ | $65.2_{\pm4.6}$ | $58.3_{\pm4.4}$ | $96.9_{\pm3.2}$ | $82.1_{\pm3.5}$ | $120.0_{\pm10.2}$ | $77.4_{\pm9.4}$ | $74.3_{\pm1.9}$ | $41.3_{\pm1.8}$ |
| GD | $10.1_{\pm0.9}$ | $2.3_{\pm0.8}$ | $19.1_{\pm0.9}$ | $3.9_{\pm0.5}$ | $61.2_{\pm3.2}$ | $16.6_{\pm0.6}$ | $69.0_{\pm2.2}$ | $46.2_{\pm12.9}$ |
| Diag | $14.3_{\pm1.3}$ | $5.2_{\pm1.2}$ | $27.7_{\pm2.2}$ | $9.1_{\pm2.7}$ | $72.6_{\pm3.2}$ | $17.9_{\pm1.8}$ | $73.8_{\pm2.8}$ | $25.8_{\pm9.4}$ |
| **Rank-1 (ours)** | $\mathbf{7.6}_{\pm0.1}$ | $\mathbf{0.6}_{\pm0.1}$ | $\mathbf{15.4}_{\pm0.6}$ | $\mathbf{0.9}_{\pm0.3}$ | $\mathbf{50.5}_{\pm1.2}$ | $\mathbf{7.4}_{\pm1.2}$ | $\mathbf{48.5}_{\pm1.9}$ | $\mathbf{15.2}_{\pm4.8}$ |

**EWC complements generative distillation.**   We report the average FID at the final task as a comprehensive measure of performance and forgetting for each dataset and method in Table 1. Our results show that without generative distillation, EWC alone struggles to maintain a shared optimum across tasks, leading to degraded continual learning performance on all datasets. The consistently high forgetting indicates that the EWC penalty pulls the model toward the previous task's optimum while moving it away from the current task's, suggesting little to no overlap between task optima.

Using generative distillation alone substantially improves both average FID and forgetting on all datasets compared to EWC-only by encouraging the model to move toward an optima that performs well across all (including replayed) tasks. When combined with our rank-1 EWC, we observe further improvements on all datasets. On MNIST and FMNIST, catastrophic forgetting is nearly eliminated with forgetting $\mathcal{F} = 0.6_{\pm0.1}$ and $\mathcal{F} = 0.9_{\pm0.3}$, respectively. On the long horizon ImageNet-1k dataset, rank-1 EWC more than halves forgetting relative to generative distillation alone ($\mathcal{F} = 15.2_{\pm4.8}$ vs. $\mathcal{F} = 46.2_{\pm12.9}$). And across all datasets, image generation quality improves substantially, further narrowing the gap to the non-continual upper bound. In contrast, while combining diagonal EWC with distillation improves the diagonal EWC-only approach, the improvement stems largely from distillation, and performance often matches or even degrades compared to distillation alone, suggesting the ineffectiveness of the diagonal EWC constraint. Overall, these results demonstrate that rank-1 EWC effectively complements generative distillation by enforcing parameter constraints within the shared optimum that distillation provides. We provide additional analysis of low-rank Fisher estimation (ranks 1–5) via power iteration, as well as continual fine-tuning baselines, in Appendix E.4.

**Rank-1 EWC is stable and robust on long horizon tasks.**   To study the learning dynamics during continual learning on evaluated datasets, we plot the average FID at each task in Figure 4. Our results show that rank-1 EWC with distillation consistently reaches a lower average FID during learning on all datasets than distillation-only and the diagonal variant. Surprisingly, on MNIST and FMNIST, average FID even decreases as the model learns new tasks. Given the near-zero positive forgetting from Table 1, Figure 4 suggests that the generation quality on some tasks improve when learning new tasks. This improvement in generation quality on early trained tasks can also be found in CIFAR-10 where the average FID decreases at the final task. This finding implies that our rank-1 EWC approach not only effectively constrained the model updates to preserve knowledge on the old tasks, but also refining old knowledge based on new tasks.

In addition, on the long horizon ImageNet-1k dataset, both generative distillation-only and the diagonal variant begin to diverge around task 10 while the average FID only gradually increases for

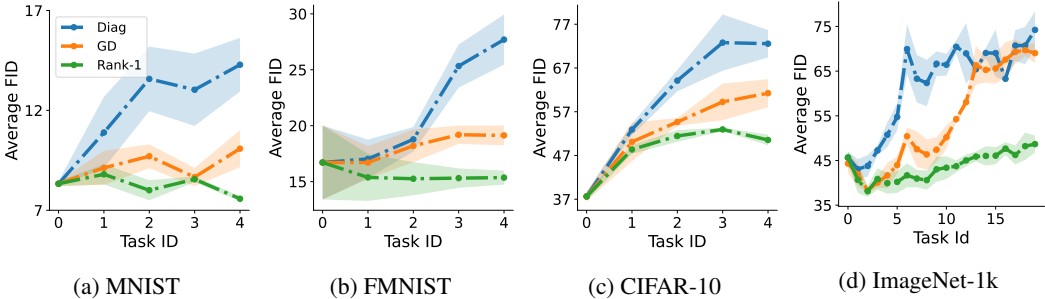

|  |  |  |  |
|---|---|---|---|
| (a) MNIST | (b) FMNIST | (c) CIFAR-10 | (d) ImageNet-1k |

Figure 4: Average FID at each task during continual learning on evaluated datasets. Standard errors are averaged over 3 random seeds.

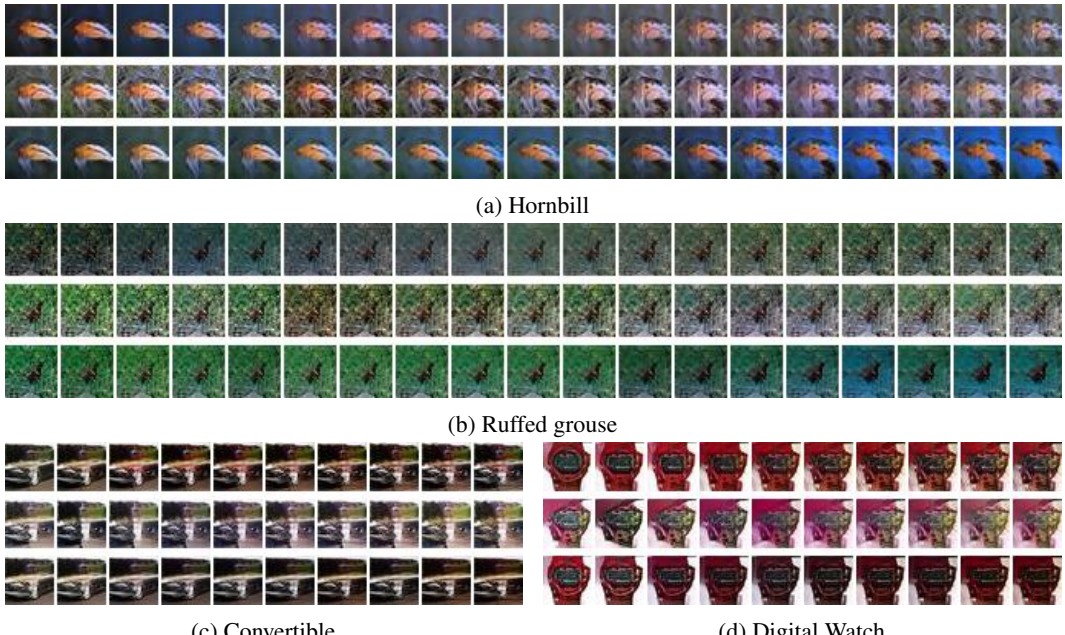

(a) Hornbill

(b) Ruffed grouse

(c) Convertible                    (d) Digital Watch

Figure 5: Examples of generated images from selected classes in ImageNet-1k over continual learning tasks. (a) Hornbill class sampled from models trained on task 1 to 19. (b) Ruffed grouse class from task 1 to 19. (c) Convertible class from task 10 to 19. (d) Digital watch class from task 10 to 19. Top row: generative distillation-only; middle row: diagonal; bottom row: rank-1.

our rank-1 approach. In particular, the generative distillation-only approach reached a plateau around task 6 to 11 before diverging. This suggests that distillation suffers from distribution shift due to errors that have accumulated from the previous imperfect replay samples, such that the model optima moves away from early tasks. We refer Appendix E.5 for additional results.

## 5.2 QUALITATIVE ANALYSIS

To visually inspect how image generation quality evolves during continual learning, we fix a label class and, after each task, sample images from that class and plot them in sequence. Figure 5 illustrates ImageNet-1k examples: the hornbill and the ruffed grouse classes sampled from models trained on task 1-19 (Figs. 5a and 5b), the convertible and the digital watch classes sampled from models trained on tasks 10–19 ( Figs. 5c and 5d). Our results show that both generative distillation-only (top row) and the diagonal variant (middle row) progressively generate a noisier sample as continual learning progresses to later tasks. On the other hand, rank-1 EWC with generative distillation maintains the sharpness of the objects. For example, both ruffed grouse and digital watch remain intact while other approaches almost distort the central object in the later tasks. Our results show that the proposed rank-1 method consistently preserves image quality across tasks, whereas images generated by the diagonal variant and generative distillation-only progressively become noisy and less recognizable.

## 6 CONCLUSION, LIMITATIONS, AND FUTURE WORK

In this paper, we investigate gradients in diffusion models and show that per-sample gradients become approximately collinear with their population mean when the SNR is low, inducing an effective rank-1 Fisher information matrix. Leveraging this structure, we hypothesized and validated that a rank-1 Fisher provides a better approximation for EWC than the commonly used diagonal Fisher approximation. When paired with generative distillation, which encourages cross-task parameter sharing that EWC assumes, our method improves continual learning performance. On class-incremental image generation, rank-1 EWC with generative distillation outperforms both generative distillation-only and diagonal Fisher EWC across MNIST, FMNIST, CIFAR-10, and ImageNet-1k in terms of lower average FID and reduced forgetting. In particular, forgetting is nearly eliminated on MNIST and FMNIST and is roughly halved on ImageNet-1k, the longest-horizon setting, relative to generative distillation-only. These findings indicate that diffusion models admit an effective rank-1 Fisher; with this better Fisher approximation, EWC complements replay by constraining replay-induced distribution drift toward a shared parameter region that supports all tasks.

**Limitations and future work.** While the main contribution of our work lies in analyzing the gradient geometry of the diffusion model's mathematical framework, an important direction for future work is to conduct a more in-depth empirical and theoretical study of Assumption 1 across multiple model architectures. Such an analysis would help clarify which architectural components (e.g., skip connections, attention mechanisms, etc.) or training regularizations (e.g., L2 norms) make Assumption 1 more or less likely to hold.

## REPRODUCIBILITY STATEMENT

We used open-source datasets and model implementations as described in Section 3.2 and Section 4, and additional experiment details, including empirical validation experiment implementations, class-incremental dataset splits, and hyper-parameters in Appendices D and E.3.

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

## A   THE USE OF LARGE LANGUAGE MODELS

We primarily use LLMs to improve wording and sentence structure, check grammar, and reorganize or document code.

## B   ADDITIONAL PROOFS

### B.1   PROOF FOR PROPOSITION 1

*Proof.* By Tweedie's identity, write $s_t^\star(x_t) = \frac{\sqrt{\bar{\alpha}_t}}{1-\bar{\alpha}_t}\mathbb{E}[x_0 \mid x_t] - x_t/(1-\bar{\alpha}_t)$. As $\bar{\alpha}_t$ decreases with timestep due to DDPM's linear noise scheduler, SNR= $\frac{\sqrt{\bar{\alpha}_t}}{1-\bar{\alpha}_t}$ decreases with $t$. And because $\frac{\sqrt{\bar{\alpha}_t}}{1-\bar{\alpha}_t}$ decreases faster than $1/(1-\bar{\alpha}_t)$, $s_t^\star(x_t) \approx -x_t/(1-\bar{\alpha}_t)$. □

### B.2   PROOF FOR PROPOSITION 2

*Proof.* Let $g(\theta; x_t) = \nabla_\theta \mathcal{L}_{\text{DSM}}(\theta; x_t)$, $\mu(\theta) = \mathbb{E}_{x'_t \sim q_t}[g(\theta; x'_t)]$. By Proposition 1, $s^\star(x_t) \approx x_t\gamma_t$, where $\gamma_t = -1/(1-\bar{\alpha}_t)$ is a scalar independent of $x_t$. Near convergence, the denoising network approximates the true score function: $s_\theta(x_t, t) \approx s^\star(x_t) \approx x_t\gamma_t$. By Assumption 1, $s_\theta(x_t, t)$ takes a linear form as $s_\theta(x_t, t) = A_\theta x_t$. Substitute back to $g(\theta; x_t)$ :

$$g(\theta; x_t) = \nabla_\theta \left( \frac{1-\bar{\alpha}_t}{2} \| s_\theta(x_t, t) - s^\star(x_t) \|_2^2 \right) \tag{8}$$

$$= \nabla_\theta \left( \frac{1-\bar{\alpha}_t}{2} \| A_\theta x_t - x_t\gamma_t \|_2^2 \right) \tag{9}$$

$$= (1-\bar{\alpha}_t)(A_\theta x_t - x_t\gamma_t I)x_t^\top \tag{10}$$

$$= (1-\bar{\alpha}_t)\|x_t\|^2 (A_\theta - \gamma_t I). \tag{11}$$

Let $c(x_t) = (1-\bar{\alpha}_t)\|x_t\|^2$ be a scalar function dependent on $x_t$, and $v = A_\theta - \gamma_t I$ be a vector in parameter space doesn't depend on $x_t$. Then $g(\theta; x_t) = c(x_t)v$. By the definition of collinearity, $g(\theta; x_t)$ collinear with $v$ for any $x_t \sim q_t$. And since per-sample gradient collinear with each other, it collinear with the population mean $\mu(\theta)$. □

## C    ADDITIONAL DISCUSSION ON ASSUMPTION 1

The theoretical rationale behind Assumption 1 comes from the analysis of autoencoders. Let $f_\theta$ be an autoencoder trained with the standard MSE objective to approximate the identity mapping, i.e., $f_\theta(x) \approx x$. As stated in Proposition 1, in the late diffusion steps the denoising network behaves approximately like such an autoencoder $f_\theta$.

Vincent et al. (2010) report that an *autoencoder with MSE* is doing PCA-like operation when there is no nonlinearity. When nonlinearities are introduced, they argue that the learned representation is still "likely" to lie in a PCA subspace. For alternative reconstruction losses such as cross-entropy, Vincent et al. (2010) note that the autoencoder is no longer the same as that of PCA and will likely learn different features. In our setting, this suggests that the observed approximately rank-1 behavior is primarily a consequence of the specific DDPM parameterization: a non-parameterized, variance-preserving forward process that reduces the ELBO to a simple MSE objective. In particular, this rank-1 behavior need not hold when the training objective is not purely MSE.

To illustrate this, we also experimented with a VAE (Kingma & Welling, 2014) trained with an MSE + KL objective. To quantify rank-1 dominance, we measure the percentage of total variance explained by the largest eigenvalue of the empirical Fisher information matrix:

$$\frac{\lambda_1}{\sum_i \lambda_i}.$$

When analyzing the gradients of this VAE, we found that the top eigenvalue accounts for only about $50\%$ of the total variance. In contrast, for DDPM in late timesteps, the top eigenvalue explains roughly $99\%$ of the variance.

Moreover, when we down-weight the KL term by a factor of $10^{-3}$, the top eigenvalue in the VAE setting explains about $85\%$ of the variance, suggesting that the near rank-1 behavior is closely tied to the autoencoding regime with an MSE-dominated loss.

This is consistent with our theoretical analysis: even if $f_\theta(x) \approx A_\theta x$ as it works in a PCA subspace for some affine parameter $A_\theta$, the gradient of a combined MSE + KL objective cannot in general be factored as $\nabla_\theta \mathcal{L}(x) = c(x)\, v$, where $c(x)$ is a scalar depending on $x$ and $v$ is a parameter-space vector independent of $x$. Such a factorization is precisely the condition for gradient collinearity established in Appendix B.2.

Although a rigorous mathematical proof of this behavior for large-scale nonlinear autoencoder architectures is currently out of reach, we assume that the skip connections in the U-Net (Ronneberger et al., 2015) make it even more likely that the large-scale network effectively operates in such a PCA-like subspace (Ghorbani et al., 2019). Intuitively, these skip connections allow information to bypass some nonlinear activations, thereby preserving approximately linear structure in the mapping, in line with the observations of Vincent et al. (2010).

## D    ADDITIONAL EMPIRICAL RESULTS

This section provides implementation details for the empirical study presented in the main text.

We use the label-conditioned `UNet` implementation from *HuggingFace* as the denoising backbone. The base model consists of three ResNet blocks per downsampling stage, each with 16 output channels and two layers per block.

Training is performed with a batch size of 128 and learning rate $2 \times 10^{-4}$ using the Adam optimizer (Kingma & Ba, 2017). Each task is trained for 200 epochs on the full dataset. On MNIST (Lecun et al., 1998), training the base model required approximately 97 minutes on a single Nvidia A100 (40GB) GPU.

### D.1    ADDITIONAL MODEL VARIANTS

To assess whether the observed Fisher structure depends on model capacity, we trained two reduced variants of the UNet under identical hyperparameters:

1. **Small-1:** one ResNet block with 16 output channels and a single layer per block;

2. **Small-3:** three ResNet blocks with 16 output channels each, but only one layer per block.

Both models were trained for 200 epochs per task with batch size 128 and learning rate $2 \times 10^{-4}$. On MNIST, training required $\sim 56$ minutes for Small-1 and $\sim 78$ minutes for Small-3, compared to 97 minutes for the base model.

While all variants display the same qualitative trends—eigenspectrum decay (Figs. 6a and 6c) and dominance of a single eigenvalue (Figs. 6b and 6d)—we observe that the empirical rank-1 behavior becomes more pronounced as model size increases. This suggests that our theoretical predictions are not artifacts of small networks, but rather strengthen with scale. Theoretically, this follows from our optimality assumption: larger models are expected to achieve solutions closer to the optimal point, thereby aligning more closely with the conditions under which the Fisher reduces to a rank-1 structure. Consequently, one should expect the rank-1 Fisher approximation to hold even more robustly in larger, real-world diffusion models.

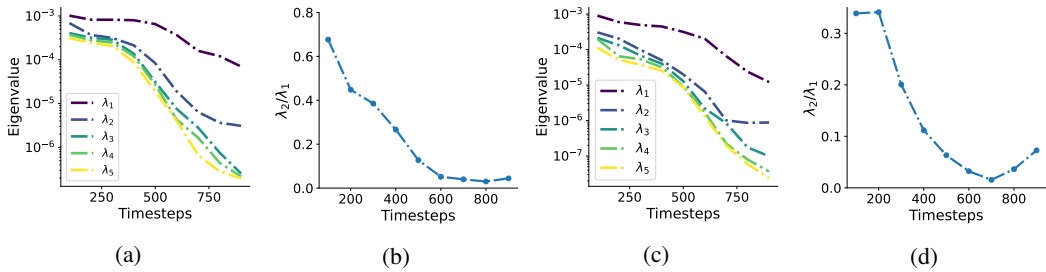

(a)       (b)       (c)       (d)

Figure 6: Eigenvalue analysis of the empirical Fisher for reduced UNet variants. Panels (a) and (b) show the top 5 eigenvalues (log-scale) and eigengap ratios $r_t = \lambda_2 / \lambda_1$ across timesteps for the **Small-1** model, while panels (c) and (d) report the same quantities for the **Small-3** model. In both cases, the Fisher remains nearly rank-1, with the leading eigenvalue $\lambda_1$ dominating and the eigengap widening as model size increases.

# E    ADDITIONAL CONTINUAL LEARNING RESULTS

## E.1    ADDITIONAL IMPLEMENTATION DETAILS

We use the label-conditioned UNet implementation from *Huggingface* as our denoising network. The UNet employs three ResNet blocks with 16 output channels in each downsampling block. For training, we use a batch size of 128 and a learning rate of $2 \times 10^{-4}$ with the Adam optimizer (Kingma & Ba, 2017), and train for 200 epochs per task for each dataset.

Training was performed on a single Nvidia A100 40GB GPU, taking a total of 157 minutes.

## E.2    ADDITIONAL IMPLEMENTATION DETAILS FOR CONTINUAL LEARNING EXPERIMENTS

We employ a label-conditioned UNet implementation from *HuggingFace* as our denoising network, retaining the default hyper-parameters. The architecture consists of four ResNet blocks, with 128 output channels in the first down-sampling block and 256 output channels in the remaining blocks. For sampling, we use a DDIM scheduler with 50 sampling steps and 1000 noising steps.

We summarize our training hyperparameters in Table 2. Unless otherwise stated, the same settings are used across all datasets and tasks. We also provide the average runtime (including FID evaluation) for each dataset and method in Table 3.

## E.3    ADDITIONAL DATASET DETAILS

All datasets are resized or padded to $32 \times 32$ for consistency. Table 4 summarizes their configurations.

**MNIST.** A dataset of handwritten digits (0–9) with 60,000 training and 10,000 test images. Each task contains two digit classes.
Task splits: $T_1 = \{0, 1\}$, $T_2 = \{2, 3\}$, $T_3 = \{4, 5\}$, $T_4 = \{6, 7\}$, $T_5 = \{8, 9\}$.

Table 2: Training configurations used across datasets.

| Setting | Value |
|---|---|
| Optimizer | Adam (Kingma & Ba, 2017) |
| Learning Rate | $2 \times 10^{-4}$ |
| Batch Size | 128 |
| Training Epochs | 200 per task |
| EWC Penalty Weight | 15000 |
| Replay Buffer (ImageNet-1k) | 1300 images per class |
| Replay Buffer (others) | 5000 images per class |

Table 3: Average training runtime (hours) and GPU used per dataset/method.

| Dataset | Method | Hours | GPU |
|---|---|---|---|
| MNIST | Diag | $\sim 5$ | NVIDIA L40S |
| MNIST | Rank-1 | $\sim 5$ | NVIDIA L40S |
| MNIST | GR | $\sim 8$ | NVIDIA L40S |
| MNIST | Diag + GR | $\sim 9$ | NVIDIA L40S |
| MNIST | Rank-1 + GR | $\sim 9$ | NVIDIA L40S |
| FMNIST | Diag | $\sim 5$ | NVIDIA L40S |
| FMNIST | Rank-1 | $\sim 5$ | NVIDIA L40S |
| FMNIST | GR | $\sim 8$ | NVIDIA L40S |
| FMNIST | Diag + GR | $\sim 9$ | NVIDIA L40S |
| FMNIST | Rank-1 + GR | $\sim 9$ | NVIDIA L40S |
| CIFAR-10 | Diag | $\sim 4$ | NVIDIA L40S |
| CIFAR-10 | Rank-1 | $\sim 4$ | NVIDIA L40S |
| CIFAR-10 | GR | $\sim 7$ | NVIDIA L40S |
| CIFAR-10 | Diag + GR | $\sim 8$ | NVIDIA L40S |
| CIFAR-10 | Rank-1 + GR | $\sim 8$ | NVIDIA L40S |
| ImageNet-1k | Diag | $\sim 25$ | NVIDIA H200 |
| ImageNet-1k | Rank-1 | $\sim 24$ | NVIDIA H200 |
| ImageNet-1k | GR | $\sim 64$ | NVIDIA H200 |
| ImageNet-1k | Diag + GR | $\sim 71$ | NVIDIA H200 |
| ImageNet-1k | Rank-1 + GR | $\sim 70$ | NVIDIA H200 |

**Fashion MNIST.**
A dataset of 10 grayscale clothing categories (e.g., shirts, shoes, bags) with 60,000 training and 10,000 test images. Each task contains two categories.
Task splits: $T_1 = \{0, 1\}$, $T_2 = \{2, 3\}$, $T_3 = \{4, 5\}$, $T_4 = \{6, 7\}$, $T_5 = \{8, 9\}$.

**CIFAR-10.**
A dataset of $32 \times 32$ RGB images across 10 classes (e.g., animals, vehicles) with 50,000 training and 10,000 test images. Each task contains two classes.
Task splits: $T_1 = \{0, 1\}$, $T_2 = \{2, 3\}$, $T_3 = \{4, 5\}$, $T_4 = \{6, 7\}$, $T_5 = \{8, 9\}$.

**ImageNet-1k (downsampled).**
A large-scale dataset with 1,000 object categories and 1.28 million training images. We use the $32 \times 32$ downsampled version for computational efficiency. Each task contains fifty classes.
Task splits: $T_1 = \{0, \ldots, 49\}$, $T_2 = \{50, \ldots, 99\}$, $\ldots$ $T_{20} = \{950, \ldots, 999\}$.

### E.4    ADDITIONAL BASELINES

### E.4.1    ABLATIONS ON LOW-RANK FISHER SURROGATES

A key practical benefit of our analysis is that it yields an efficient and scalable estimate of the leading Fisher eigen-direction using only the empirical mean gradient $\mathbb{E}[g]$, avoiding explicit formation of the Fisher and the costly iterative procedures typically required to extract top eigenvectors in high-

| Dataset | #Training Images per Task | #Tasks | Description of Each Task |
|---|---|---|---|
| MNIST (Lecun et al., 1998) | 12,000 | 5 | Generation of 2 classes of handwritten digits |
| Fashion-MNIST (Xiao et al., 2017) | 12,000 | 5 | Generation of 2 classes of fashion products |
| CIFAR-10 (Krizhevsky, 2009) | 10,000 | 5 | Generation of 2 classes of common items |
| ImageNet-1k (Chrabaszcz et al., 2017) | ~64,000 | 20 | Generation of 50 classes of ImageNet objects |

Table 4: Detailed dataset configurations and task partitions used in our experiments.

dimensional parameter spaces. To contextualize this design choice, we compare against low-rank baselines that estimate the top-$k$ Fisher subspace via stochastic power iteration and then construct rank-$k$ approximations (rank-1 through rank-5). Across both CIFAR-10 and FashionMNIST, these iterative low-rank baselines do not improve performance over our mean-gradient rank-1 estimate, and in many cases perform worse, consistent with the difficulty of reliably converging power iteration under limited compute. Moreover, increasing the approximation rank beyond 1 yields little benefit, supporting the view that the curvature is already well-captured by the dominant direction in this regime.

Table 5: Low-rank ablations using rank-$k$ Fisher approximations (estimated via stochastic power iteration) versus our mean-gradient rank-1 estimate. We report final FID and average forgetting (mean $\pm$ standard error).

| | CIFAR-10 | | FashionMNIST | |
|---|---|---|---|---|
| Method | Final FID | Avg Forgetting | Final FID | Avg Forgetting |
| GD + rank-1 (ours) | $50.5 \pm 1.2$ | $7.4 \pm 1.2$ | $15.4 \pm 0.6$ | $0.9 \pm 0.3$ |
| GD + rank-2 | $57.5 \pm 2.6$ | $15.9 \pm 1.8$ | $18.1 \pm 1.6$ | $3.8 \pm 1.4$ |
| GD + rank-3 | $60.4 \pm 5.4$ | $17.7 \pm 5.0$ | $16.8 \pm 0.9$ | $2.9 \pm 0.6$ |
| GD + rank-4 | $55.7 \pm 3.4$ | $13.0 \pm 2.0$ | $17.0 \pm 1.2$ | $2.9 \pm 1.2$ |
| GD + rank-5 | $55.6 \pm 2.3$ | $12.6 \pm 1.6$ | $17.1 \pm 1.2$ | $2.9 \pm 1.0$ |

### E.4.2 CONTINUAL FINE-TUNING WITHOUT REGULARIZATION

We additionally report the performance of plain continual fine-tuning with no regularization (i.e., setting the penalty weight to $\lambda = 0$). This baseline isolates the extent of catastrophic forgetting in the absence of any stabilization mechanism, and serves as a sanity check that our regularized baselines are not benefiting from overly conservative hyperparameters. As shown in Table 6, removing regularization yields substantially worse final generation quality and severe forgetting across datasets.

Table 6: Continual fine-tuning with no regularization ($\lambda = 0$). We report final FID and average forgetting.

| Dataset | Final FID | Avg Forgetting |
|---|---|---|
| CIFAR-10 | 115.410 | 72.496 |
| FashionMNIST | 103.955 | 88.009 |
| ImageNet32 | 265.902 | 67.864 |
| MNIST | 102.642 | 93.700 |

### E.5 CONTINUAL LEARNING CURVES FOR EACH TASK AND DATASET

We provide a comprehensive set of FID evaluation plots for all tasks across the datasets considered in our experiments. These plots illustrate the model's performance on individual tasks and complement the quantitative results presented in the main paper.

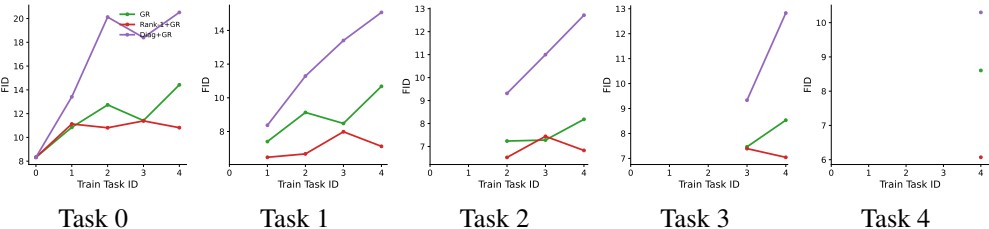

Figure 7: FID plots for MNIST (generative replay).

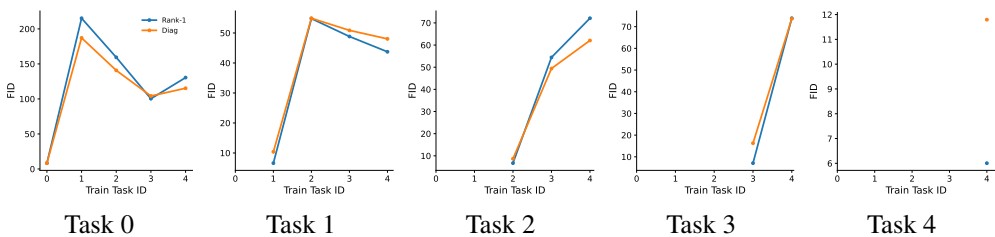

Figure 8: FID plots for MNIST (non-generative replay).

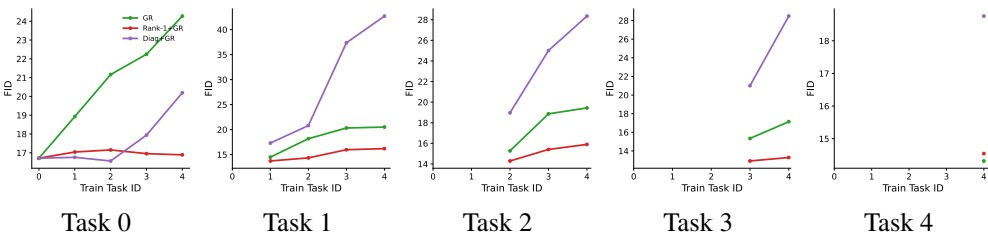

Figure 9: FID plots for Fashion-MNIST (generative replay).

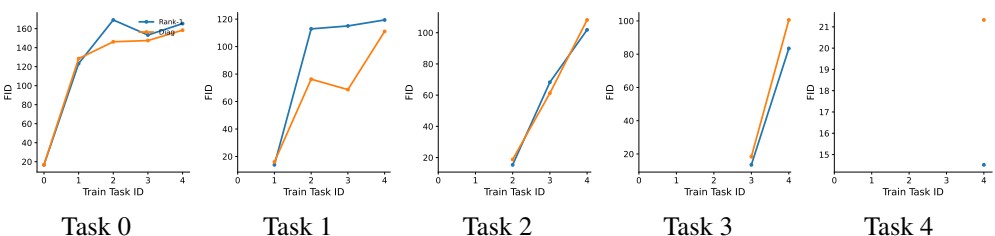

Figure 10: FID plots for Fashion-MNIST (non-generative replay).

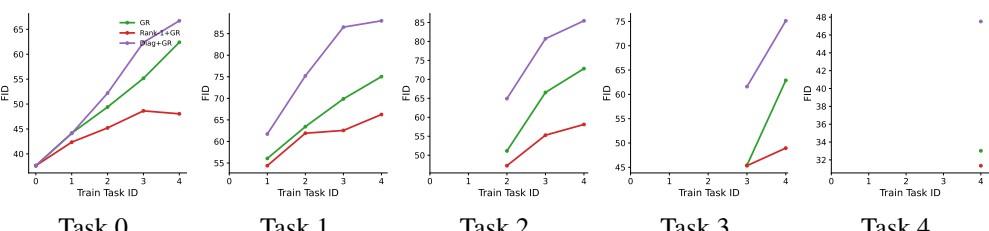

Figure 11: FID plots for CIFAR-10 (generative replay).

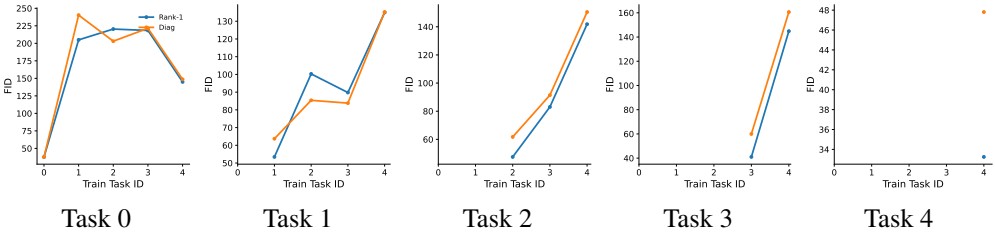

Figure 12: FID plots for CIFAR-10 (non-generative replay).

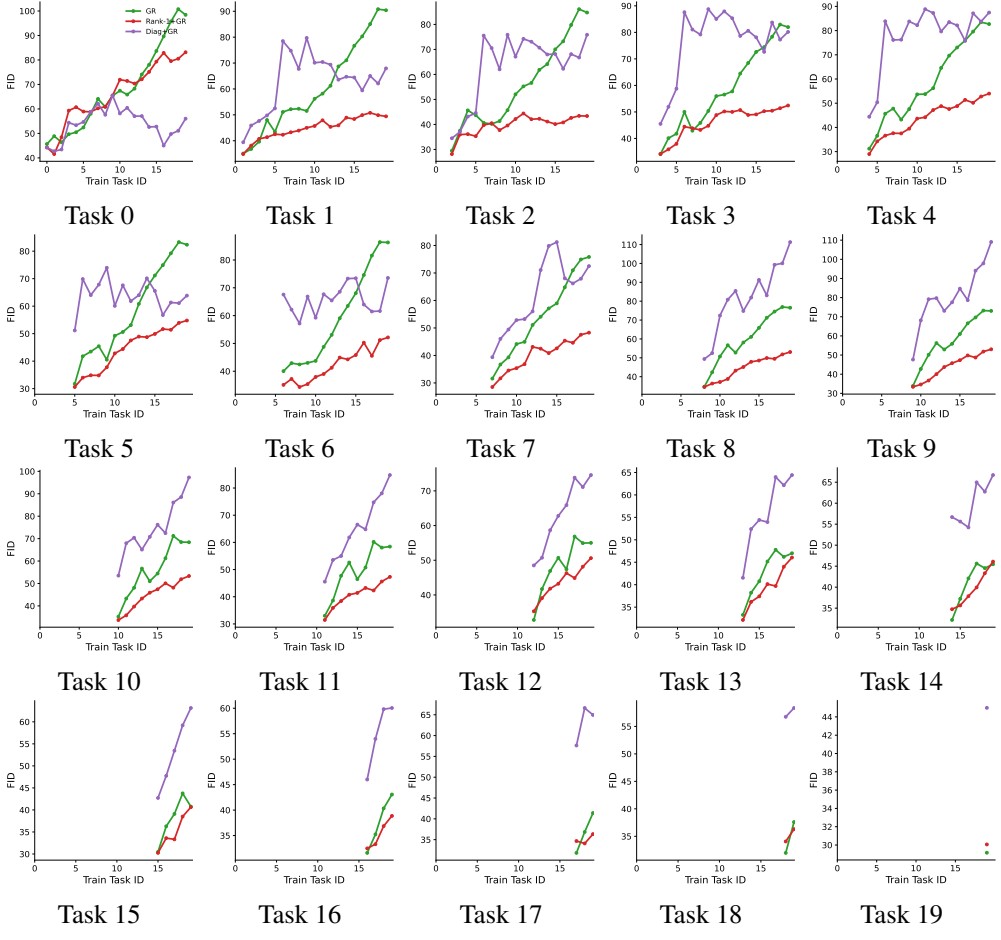

Figure 13: FID plots for Imagenet-1k (generative replay).

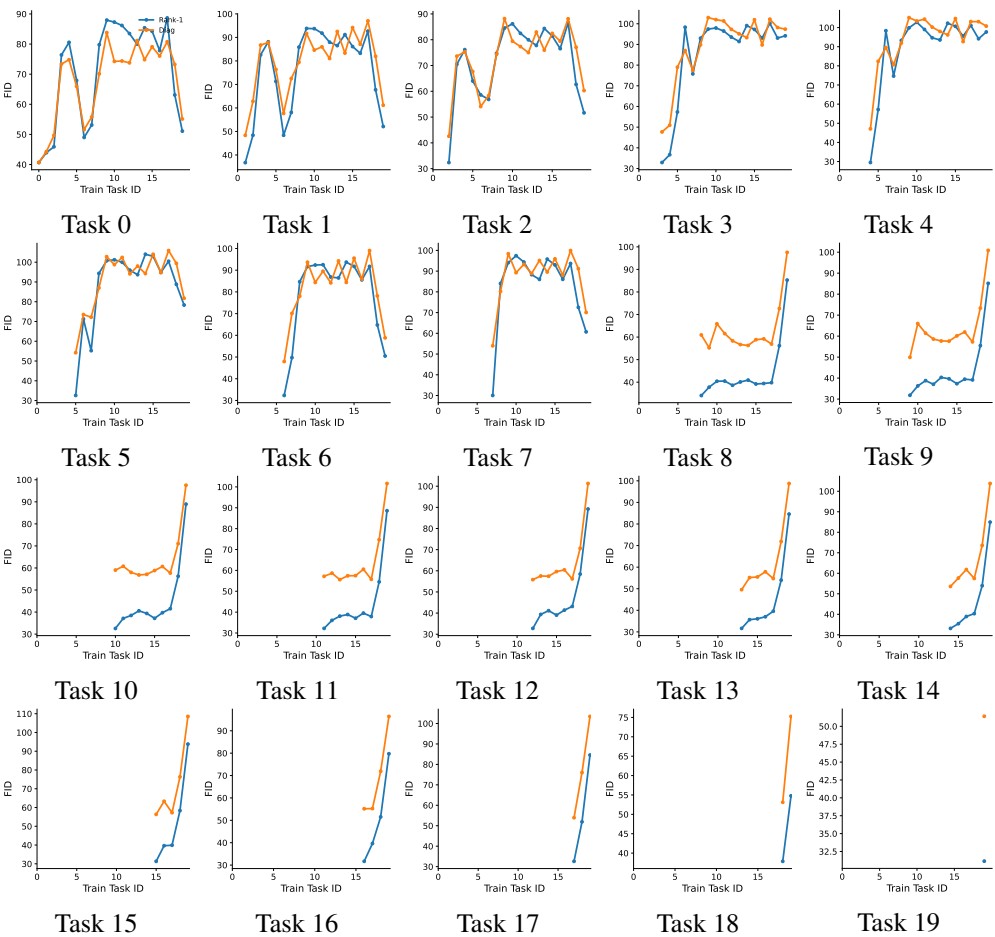

Figure 14: FID plots for Imagenet-1k (non-generative replay).

