# OpenReview forum: "Avoid Catastrophic Forgetting with Rank-1 Fisher from Diffusion Models"
_ICLR.cc/2026/Conference — ICLR 2026 Poster_

### Official Review · Reviewer_HQm2 · 2025-10-28

**Soundness:** 3
**Presentation:** 3
**Contribution:** 3
**Rating:** 6
**Confidence:** 4

**Summary:**

This paper studies the gradient geometry of diffusion models in low-SNR timesteps and argues that per-sample gradients become nearly collinear, making the empirical Fisher effectively rank-1. Building on this, the authors propose a rank-1 EWC penalty (cheap to compute) and pair it with generative distillation for continual image generation. Across MNIST, FMNIST, CIFAR-10, and downsampled ImageNet-1k (32×32), the method improves average FID and notably reduces forgetting versus diagonal-Fisher EWC and distillation alone (e.g., near-zero forgetting on MNIST/FMNIst and roughly halved on ImageNet-1k). Theory (Propositions/Theorem) and empirical analyses (eigenspectra, cosine similarities) support the rank-1 claim.

**Strengths:**

Clear theory → practice bridge: formal propositions/theorem (low-SNR → gradient collinearity → rank-1 Fisher) with convincing empirical validation (eigenvalue dominance, Frobenius error vs. diagonal).

Simple & efficient: rank-1 EWC captures dominant curvature at cost comparable to diagonal EWC; easy to add to standard UNet diffusion setups.

Consistent gains: with generative distillation, improves AFID and reduces forgetting on four datasets; forgetting ≈ 0 on MNIST/FMNIst and ~halved on ImageNet-1k.

Robust learning dynamics: smoother curves over long horizons; qualitative samples maintain object sharpness compared to baselines that drift.

**Weaknesses:**

Scope/scale limits: experiments are at 32×32 resolution (even for ImageNet-1k) and on UNet backbones; no results on larger resolutions or Transformer/DiT diffusion models, which the paper itself flags as future work.

Heavy reliance on distillation: EWC alone underperforms; the strongest results require rank-1 EWC + generative distillation, so the standalone benefit of the rank-1 penalty is limited.

**Questions:**

see weakness

---

> ### Author Response · Authors · 2025-11-18
> **We appreciate that you find our paper clear and easy to follow, and that you view our approach as simple and effective.**
>
> # "...experiments are at 32×32 resolution..."
>
> We emphasize that the primary goal of our ImageNet experiments is to examine whether our rank-1 approximation remains effective in a long-horizon continual learning setting. Specifically, we seek to test whether the rank-1 Fisher approximation is sufficiently accurate that errors from early tasks do not accumulate catastrophically over many subsequent tasks, while effectively mitigating distribution drift induced by replay.
>
> To make these long-horizon experiments computationally feasible, we adopt Downsampled ImageNet (Chrabaszcz et al., 2017) as a budget-friendly alternative to the full-resolution dataset. The ImageNet32x32 **preserves key training characteristics** of the original ImageNet
> while substantially reducing computational cost, which in turn enables faster experimental iteration and more extensive evaluation of our method.
>
>
> # "...on larger resolutions or Transformer/DiT diffusion models..."
>
> As we explained in our response to **reviewer AxKy**, the theoretical rationale behind our approach (and Assumption 1) is that an autoencoder trained with an MSE objective is ``likely'' to operate in a PCA-like subspace even in the presence of nonlinearities (Vincent et al., 2010), which in turn
> implies that its Fisher information matrix is nearly rank-1. Assumption 1 further posits that a U-Net
> is even more likely to operate in such a PCA-like subspace due to its skip connections, which allow information to bypass some nonlinear activations.
>
> While this rationale is broadly architecture-agnostic (only assumes autoencoding with MSE loss), different architectures may exhibit
> different empirical characteristics. **For DiT, our preliminary investigation indicates that it
> requires a substantially larger model size to converge reliably.**
>
> On FashionMNIST, we first trained a DiT with a parameter count comparable to our U-Net model. After the first task (with no continual learning applied yet), this DiT achieved an FID of $27.0$ on
> FashionMNIST, whereas the U-Net achieved an FID of $16.5$, suggesting that the DiT failed to fully
> converge at this model size. When we roughly doubled the number of DiT parameters, the DiT reached a similar FID to the U-Net on the first task. **Using this larger DiT, we then ran the continual learning experiment with our rank-1 method and observed that it outperformed the generative distillation baseline (FID $20.5$ vs. $29.3$), suggesting that our approach also generalizes to DiT, albeit requiring a larger model to do so.**
>
> While the main contribution of our work lies in analyzing the gradient geometry of the diffusion model’s mathematical framework, we agree that an important direction for future work is to conduct a more in-depth empirical and theoretical study of Assumption 1 across multiple model architectures. Such an analysis would help clarify which architectural components (e.g., skip connections, attention mechanisms, etc.) or training regularizations (e.g., L2 norms) make assumption 1 more or less likely to hold.
>
> *Chrabaszcz, P., Loshchilov, I., & Hutter, F. (2017). A Downsampled Variant of ImageNet as an Alternative to the CIFAR datasets (Version 3). arXiv. https://doi.org/10.48550/ARXIV.1707.08819*
>
> *Pascal Vincent, Hugo Larochelle, Isabelle Lajoie, Yoshua Bengio, and Pierre-Antoine Manzagol. 2010. Stacked Denoising Autoencoders: Learning Useful Representations in a Deep Network with a Local Denoising Criterion. J. Mach. Learn. Res. 11 (3/1/2010), 3371–3408.*

---

> ### Author Response · Authors · 2025-11-18
>
> # Rely on generative distillation
>
> **We argue that relying on replay-based approaches (e.g., generative distillation) is not a limitation, but rather a way to help satisfy the core assumption behind EWC: that the optimal parameter regions for different tasks overlap (Kirkpatrick et al., 2017).**
>
> In overparameterized networks, however, the optimal parameter sets for individual tasks may become effectively disjoint, violating this assumption. **In such a regime, even an exact Fisher matrix cannot guarantee that learning on a new task will be constrained to a parameter region that is jointly optimal for all tasks, simply because no such shared region exists.**
>
> Replay on earlier tasks alleviates this issue by exposing the model to a larger, joint support over
> all (replayed) tasks, thereby encouraging the existence of overlapping optimal parameter regions. **However, replay alone suffers from distribution drift, and errors can accumulate due to imperfect generations of past data.** For example, in our experiments on long-horizon ImageNet tasks, the generative distillation (GD) baseline exhibits a sudden drop in performance after continual training on 10 tasks, consistent with such drift and error compounding.
>
> Our method instead leverages a **natural synergy:** replay is used to help satisfy EWC’s overlapping-parameter assumption, while our rank-1 EWC correction mitigates the distribution drift that arises from replay.
>
> *Kirkpatrick, J., Pascanu, R., Rabinowitz, N., Veness, J., Desjardins, G., Rusu, A. A., Milan, K., Quan, J., Ramalho, T., Grabska-Barwinska, A., Hassabis, D., Clopath, C., Kumaran, D., & Hadsell, R. (2017). Overcoming catastrophic forgetting in neural networks. PNAS Proceedings of the National Academy of Sciences of the United States of America, 114(13), 3521–3526. https://doi.org/10.1073/pnas.1611835114*

---

### Official Review · Reviewer_mxSL · 2025-11-01

**Soundness:** 3
**Presentation:** 3
**Contribution:** 3
**Rating:** 6
**Confidence:** 4

**Summary:**

This paper investigates the problem of catastrophic forgetting in the continual learning of generative diffusion models. The authors identify a key limitation of Elastic Weight Consolidation (EWC), a popular regularization method, which typically relies on a diagonal approximation of the Fisher Information Matrix (FIM) that fails to capture important parameter correlations. The paper's main contribution is a theoretical and empirical analysis showing that for diffusion models in the low signal-to-noise ratio (SNR) regime, the empirical FIM is effectively rank-1 and aligned with the mean gradient. Leveraging this insight, the authors propose a "Rank-1 EWC" penalty that is computationally efficient yet captures this dominant curvature direction. This regularizer is combined with generative distillation (a form of replay) to create a synergistic approach that encourages a shared parameter space while mitigating drift. Experiments on class-incremental image generation tasks (MNIST, FashionMNIST, CIFAR-10, and ImageNet-1k) show that the proposed method significantly reduces forgetting and improves sample quality (FID) compared to replay-only and diagonal-EWC baselines.

**Strengths:**

1. Novel and Insightful Analysis of Fisher Geometry: The core of the paper is a novel and non-obvious finding about the gradient structure of diffusion models.
2. Practical and Computationally Efficient Algorithm: The proposed Rank-1 EWC penalty, as formulated in Equation 6, is highly practical.
3. Strong Empirical Performance and Validation: The method demonstrates impressive empirical results, substantially reducing catastrophic forgetting and improving generation quality (FID) across four different benchmarks.

**Weaknesses:**

1. Heavy Reliance on Generative Distillation: A significant weakness is the method's apparent dependence on the replay component.
2. Strength of Theoretical Assumptions: The theoretical argument hinges on Assumption 1, which posits that the score network $s_{\theta}(x_t, t)$ can be approximated as a linear function of its parameters, $s_{\theta}(x_t, t) \approx x_t\theta$, in the low-SNR regime.
3. Limited Scope of Fisher Matrix Analysis: The detailed empirical analysis of the FIM in Section 3.2 is conducted on a small-scale diffusion model trained on MNIST

**Questions:**

1. The effectiveness of your Rank-1 EWC is clearly dependent on the generative distillation component. Could you elaborate on the precise role of the penalty?
2. Your theoretical analysis highlights the emergence of the rank-1 structure in the low-SNR (late timestep) regime. However, the EWC penalty in Equation 6 appears to be based on a mean gradient $\mu$ that is averaged over all timesteps. How does the inclusion of gradients from high-SNR timesteps, where the Fisher may not be rank-1, affect the validity and performance of your proposed penalty?
3. The analysis in Figure 3a shows that the mean gradients $\mu_t(\theta)$ are highly aligned across timesteps but are not perfectly collinear. This suggests that a single rank-1 approximation for the entire FIM might be discarding some useful information.

---

> ### Author Response · Authors · 2025-11-18
> **We thank you for considering our contribution novel and non-obvious.**
>
> # Rely on generative distillation
>
> **We argue that relying on replay-based approaches (e.g., generative distillation) is not a limitation, but rather a way to help satisfy the core assumption behind EWC: that the optimal parameter regions for different tasks overlap (Kirkpatrick et al., 2017).**
>
> In overparameterized networks, however, the optimal parameter sets for individual tasks may become effectively disjoint, violating this assumption. **In such a regime, even an exact Fisher matrix cannot guarantee that learning on a new task will be constrained to a parameter region that is jointly optimal for all tasks, simply because no such shared region exists.**
>
> Replay on earlier tasks alleviates this issue by exposing the model to a larger, joint support over
> all (replayed) tasks, thereby encouraging the existence of overlapping optimal parameter regions. **However, replay alone suffers from distribution drift, and errors can accumulate due to imperfect generations of past data.** For example, in our experiments on long-horizon ImageNet tasks, the generative distillation (GD) baseline exhibits a sudden drop in performance after continual training on 10 tasks, consistent with such drift and error compounding.
>
> Our method instead leverages a **natural synergy:** replay is used to help satisfy EWC’s overlapping-parameter assumption, while our rank-1 EWC correction mitigates the distribution drift that arises from replay.
>
> *Kirkpatrick, J., Pascanu, R., Rabinowitz, N., Veness, J., Desjardins, G., Rusu, A. A., Milan, K., Quan, J., Ramalho, T., Grabska-Barwinska, A., Hassabis, D., Clopath, C., Kumaran, D., & Hadsell, R. (2017). Overcoming catastrophic forgetting in neural networks. PNAS Proceedings of the National Academy of Sciences of the United States of America, 114(13), 3521–3526. https://doi.org/10.1073/pnas.1611835114*

---

> ### Author Response · Authors · 2025-11-18
>
> # "...How does the inclusion of gradients from high-SNR timesteps, where the Fisher may not be rank-1, affect the validity and performance of your proposed penalty?.."
>
> Consider Tweedie’s identity
> $s_t(x)
> = \frac{\sqrt{\bar{a}_t}}{1-\bar{a}_t}\,\mathbb{E}[x_0 \mid x] - \frac{x}{1-\bar{a}_t}.$
> As $\bar{a}_t \to 0$, the first term vanishes and we obtain $s_t(x) \to -x$. Now let us analyze
> $\mathrm{SNR}(t) \;=\; \frac{\sqrt{\bar{a}_t}}{1-\bar{a}_t}$
>  as a function of $t$. As $t$ increases, $\bar{a}_t$ decreases, and thus $\mathrm{SNR}(t)$ decreases as well, **with a sharp drop at early timesteps.** In other words, the SNR is low for most diffusion steps (cf.\ Figure 1, which shows a $1/x$-like decay).
>
> However, even at very early timesteps where the SNR is still relatively high, the U-Net behaves approximately like a denoising autoencoder $f_\theta(\tilde{x}) \approx x$, with $\tilde{x} = x + \mathcal{N}(0, I)$, and the empirical Fisher of the gradients remains low-rank. For example, in Figure 3c we observe that the ratio $\lambda_2 / \lambda_1 < 0.4$ at timestep $t = 100$, indicating that our cheap rank-1 approximation still captures a large portion of the curvature. In contrast, a diagonal approximation captures almost no variance at any timestep (see Figure 3d). **Thus, the rank-1 approximation can capture most of the curvature at the majority of diffusion timesteps.**
>
> In practice, instead of approximating the Fisher at each timestep separately, we average across timesteps (Figure 3a; Section 3.3). This temporal averaging dilutes the residual higher-rank structure from early timesteps and is dominated by the nearly rank-1 gradients from the remaining timesteps. Empirically, while the top eigenvalue explains about $99\\%$  of the total variance at timestep $t = 900$, the top eigenvalue of the Fisher matrix averaged over timesteps still explains approximately $86\\%$ of the total variance.
>
> We also refer to our discussion with **reviewer AxKy** for additional details.

---

> ### Author Response · Authors · 2025-11-18
>
> # "...This suggests that a single rank-1 approximation for the entire FIM might be discarding some useful information...."
>
> While mean gradients highly aligned across timesteps despite not being perfectly collinear, our rank-1 approach remain sufficient due to the following two observations:
>
> - The Fisher information is nearly rank-1 at most diffusion steps and **remains low-rank at early timesteps**, so that when averaging across timesteps, the overall Fisher is still dominated by a rank-1 structure.
> - We additionally ran experiments with low-rank baselines for ranks $1–5$ using power iteration. Our results show that adding more ranks did not necessarily improve continual learning performance. We refer to our discussion with **reviewer rvLS** for additional details.

---

### Official Review · Reviewer_AxKy · 2025-11-01

**Soundness:** 3
**Presentation:** 3
**Contribution:** 3
**Rating:** 6
**Confidence:** 4

**Summary:**

This paper concentrates on the continual learning of diffusion models with an improved EWC method (i.e., rank-1 EWC). The main contributions are summarized as:

a. This paper provides both theoretical and empirical characterizations of Fisher information geometry in diffusion models, showing that low SNR induces a near rank-1 Fisher aligned with the mean gradient.

b. This paper proposes a practical rank-1 EWC penalty that is as cheap as a diagonal penalty but captures more curvature information for diffusion models.

c. This paper provides extensive experiments to prove the effectiveness of the proposed method.

**Strengths:**

a. This paper is well written and easy to follow.

b. This paper is technically solid.

c. It is interesting and significant to prove that the empirical Fisher matrix of diffusion models is effectively rank-1.

**Weaknesses:**

I am generally satisfied with the content presented in the paper; however, I still have the following concerns:

a. I believe that the most critical premise supporting this paper is Assumption 1. However, its explanation is overly intuitive and lacks rigor. Please use a concrete example and demonstrate, from a mathematical perspective, how this example is connected to Assumption 1. This is the major concern.

b. The Theorem considers the low SNR region (i.e, at later diffusion timesteps). However, SNR will be high at most diffusion timesteps. How does the paper account for this situation?

c. How to determine that a SNR is low?

**Questions:**

a. In Section 3, it is mentioned that "This is plausible in practice because a trivial solution for UNet is to directly route the inputs to the output due to the skip connections". Please theoretically discuss Assumption 1 according to this example in details.

---

> ### Author Response · Authors · 2025-11-18
> **We appreciate that you find our work interesting and significant**
>
> # Assumption 1
> **we would like to emphasize that Assumption 1 is not the most critical premise in our argument, a more critical premise is DDPM's simplified MSE traning objective.**
>
> The theoretical rationale behind Assumption 1 comes from the analysis of autoencoders.
> Let $f_\theta$ be an autoencoder trained with the standard MSE objective to approximate
> the identity mapping, i.e., $f_\theta(x) \approx x$. As stated in Proposition 1, in the
> late diffusion steps the denoising network behaves approximately like such an
> autoencoder $f_\theta$.
>
> Vincent et al. (2010) report that an *autoencoder with MSE* is doing PCA-like operation when there is no nonlinearity. When nonlinearities are introduced, they argue that the learned representation is still ``likely'' to lie in a PCA subspace.
> For alternative reconstruction losses such as cross-entropy, Vincent et al. (2010) note that the autoencoder ‘’… is no longer the same as that of PCA and will likely learn different features….'' **In our setting, this suggests that the observed approximately rank-1 behavior is primarily a consequence of the specific DDPM parameterization: a non-parameterized, variance-preserving forward process that reduces the ELBO to a simple MSE objective.** In particular, this rank-1 behavior need not hold when the training objective is not purely MSE.
>
> To illustrate this, we also experimented with a VAE trained with an $MSE+KL$ objective. To quantify rank-1 dominance, we measure the percentage of total variance explained by the largest eigenvalue of the empirical Fisher information matrix:  $\frac{\lambda_1}{\sum_{i} \lambda_i}.$
> When analyzing the gradients of this VAE, we found that the top eigenvalue accounts for only about $50\\%$ of the total variance. In contrast, for DDPM in late timesteps, the top eigenvalue explains roughly $99\\%$  of the variance.
>
> Moreover, when we down-weight the KL term by a factor of $10^{-3}$, the top eigenvalue in the VAE
> setting explains about $85\\%$ of the variance, suggesting that the near rank-1 behavior is closely tied to the autoencoding regime with an MSE-dominated loss.
>
> This is consistent with our theoretical analysis: even if $f_{\theta}(x) \approx A_{\theta} x$ as it works in a PCA subspace for some affine parameter $A_{\theta}$, the gradient of a combined $MSE+KL$ objective cannot in general be factored as $\nabla_{\theta} \mathcal{L}(x) = c(x)v,$ where $c(x)$ is a scalar depending on $x$ and $v$ is a parameter-space vector independent of $x$. Such a factorization is precisely the condition for gradient collinearity established in Appendix B.2.
>
> Although a rigorous mathematical proof of this behavior for large-scale nonlinear
> autoencoder architectures is currently out of reach, we assume that the skip
> connections in the U-Net make it even more likely that the network effectively operates
> in such a PCA-like subspace. **Intuitively, these skip connections allow information to
> bypass some nonlinear activations, thereby preserving approximately linear structure in
> the mapping, in line with the observations of Vincent et al. (2010).** Again, Assumption 1 depends more on autoencoding with MSE loss than the network architecture. We refer to our discussion with **reviewer HQm2** regarding the Diffusion Transformer architecture.
>
> **We thank the reviewer again for initiating this interesting discussion. We will refine the rationale behind Assumption 1 in the revised version to incorporate the above clarifications.**
>
>
> *Pascal Vincent, Hugo Larochelle, Isabelle Lajoie, Yoshua Bengio, and Pierre-Antoine Manzagol. 2010. Stacked Denoising Autoencoders: Learning Useful Representations in a Deep Network with a Local Denoising Criterion. J. Mach. Learn. Res. 11 (3/1/2010), 3371–3408.*

---

> ### Author Response · Authors · 2025-11-18
>
> # What happen to intermediate/early timesteps?
>
> Consider Tweedie’s identity
> $s_t(x)
> = \frac{\sqrt{\bar{a}_t}}{1-\bar{a}_t}\,\mathbb{E}[x_0 \mid x] - \frac{x}{1-\bar{a}_t}.$
> As $\bar{a}_t \to 0$, the first term vanishes and we obtain $s_t(x) \to -x$. Now let us analyze
> $\mathrm{SNR}(t) = \frac{\sqrt{\bar{a}_t}}{1-\bar{a}_t}$
>  as a function of $t$. As $t$ increases, $\bar{a}_t$ decreases, and thus $\mathrm{SNR}(t)$ decreases as well, **with a sharp drop at early timesteps.** In other words, the SNR is low for most diffusion steps (cf. Figure 1, which shows a $1/x$-like decay).
>
> However, even at very early timesteps where the SNR is still relatively high, the U-Net behaves approximately like a denoising autoencoder $f_\theta(\tilde{x}) \approx x$, with $\tilde{x} = x + \mathcal{N}(0, I)$, and the empirical Fisher of the gradients remains low-rank. For example, in Figure 3c we observe that the ratio $\lambda_2 / \lambda_1 < 0.4$ at timestep $t = 100$, indicating that our cheap rank-1 approximation still captures a large portion of the curvature. In contrast, a diagonal approximation captures almost no variance at any timestep (see Figure 3d). **Thus, the rank-1 approximation can capture most of the curvature at the majority of diffusion timesteps.**
>
> In practice, instead of approximating the Fisher at each timestep separately, we average across timesteps (Figure 3a; Section 3.3). This temporal averaging dilutes the residual higher-rank structure from early timesteps and is dominated by the nearly rank-1 gradients from the remaining timesteps. Empirically, while the top eigenvalue explains about $99\\%$  of the total variance at timestep $t = 900$, the top eigenvalue of the Fisher matrix averaged over timesteps still explains approximately $86\\%$ of the total variance.

---

> ### Author Response · Authors · 2025-11-18
>
> # How to determine that a SNR is low
>
> While there is no universally agreed-upon threshold for what constitutes a ``low'' SNR,
> Figure 1 (MSE between the U-Net’s input and its rescaled output across timesteps)
> empirically illustrates the effect of the $\frac{\sqrt{\bar{a}_t}}{1-\bar{a}_t}\,\mathbb{E}[x_0 \mid x]$ term in Tweedie’s identity.
> As the timestep $t$ increases, this term shrinks, and the MSE between the input and the
> rescaled output correspondingly decreases. For example, in our experiments with a small U-Net
> (whose Fisher matrix of size  n_params $\times$ n_params can be stored in GPU memory), the MSE
> reaches approximately $0.1$ around timestep $t \approx 400$, suggesting that the SNR is already
> quite low at that point.
>
> In principle, one could define a ``low SNR regime'' by imposing a hard threshold on this MSE
> or on the SNR value itself, but this would introduce an additional hyperparameter that may need
> to be tuned for each experiment and each model size. Instead, in practice, we leverage two empirical observations:
>
> (i) the SNR decays sharply with $t$, and
>
> (ii) the Fisher remains strongly low-rank even at early
> timesteps. Consequently, rather than selecting a specific cutoff, we simply average the Fisher
> matrices across timesteps, which naturally emphasizes the dominant, nearly rank-1 structure that
> emerges over the course of the diffusion process.

---

### Official Review · Reviewer_rvLS · 2025-11-04

**Soundness:** 3
**Presentation:** 3
**Contribution:** 3
**Rating:** 6
**Confidence:** 3

**Summary:**

The paper studies the gradient geometry of diffusion models and argues that, at low SNR timesteps, per-sample gradients align strongly with their mean, making the empirical Fisher approximately rank-1. Building on this, the authors propose a rank-1 EWC penalty that is as cheap as the diagonal approximation but better aligned with the dominant curvature direction. They pair it with diffusion-based generative distillation/replay. On class-incremental image generation (MNIST, FMNIST, CIFAR-10, downsampled ImageNet-1k), the method improves AFID and reduces forgetting versus diagonal-EWC and replay alone; forgetting is nearly eliminated on MNIST/FM and roughly halved on ImageNet-1k.

**Strengths:**

- The motivation example is well positioned, the story of the paper is clear and easy to follow.
- Clean connection from low SNR → gradient collinearity → rank-1 Fisher, with both theory and measurement
- Rank-1 EWC that is drop-in and cost-comparable to diagonal while capturing dominant curvature.
- Consistent AFID/forgetting improvements; notably near-zero forgetting and significant improvement on different benchmarks.

**Weaknesses:**

- Ablations missing/limited: (i) k-rank (>1) Fisher vs rank-1 vs diagonal; (ii) Sensitivity to λ, μ-estimation schedule, and the SNR/timestep sampling strategy; (iii) replay buffer size; teacher choice (EMA vs last checkpoint) in distillation.

- No wall-clock/memory comparisons vs Diag-EWC and low-rank baselines (e.g., diagonal + small K-FAC block, diagonal + momentum/EMA Fisher smoothing).

- Main long-horizon test uses 32×32 ImageNet-1k; higher-res or domain/task-incremental setups would strengthen claims.

- I suspect that other baselines are not well tuned. Figure 4 shows that the proposed method is consistently better than others in adapting to new task, which means the method can obtain both stability and plasticity overall. I am wondering how the result would look like if λ=0 (no regularization).

**Questions:**

- Is this the first work about continual learning for diffusion model? If it is not, I expect the author to compare against other related work. The main table result looks like an ablation studies for Fisher approximation approaches.

- Can you conduct ablation studies on rank-k (k=2–8) variants to test diminishing returns beyond rank-1?

---

> ### Author Response · Authors · 2025-11-18
> **We thank the reviewer for considering our work well-written and well-motivated.**
>
> # Ablation on low-rank baselines.
>
> - We would like to clarify that one of our main contributions is not only demonstrating that the Fisher information matrix is effectively **rank-1**, but also providing an **efficient and scalable method** for computing its top eigenvector by using the empirical mean gradient **E[g]**.
> - In general, computing the top eigenvectors of the Fisher requires **stochastic power iteration**, which is expensive due to (1) the large number of training samples and (2) the slow numerical convergence of the power method in high-dimensional parameter spaces. To illustrate this challenge, we ran experiments on CIFAR-10 and FMNIST using **4 iterations** of stochastic power iteration to estimate rank-1 through rank-5 approximations. These approximations performed **worse than our mean-gradient rank-1 estimate**, precisely because convergence is difficult even with 4× the compute. We also ran experiments with **8 iterations** yielding similiar results. Given the high dimensionality, it is hard to quantify how many iterations it will take for convergence.
> - Moreover, even when increasing the Fisher rank from 2 to 5, we observe **little change in the final FID score**. This further supports our claim that the Fisher is effectively rank-1 in this setting and that our **efficient rank-1 computation is both sufficient and preferable**.
>
> | **Method** | **CIFAR-10** |  | **FashionMNIST** |  |
> | --- | --- | --- | --- | --- |
> |  | Final FID | Avg Forgetting | Final FID | Avg Forgetting |
> | **GD + rank-1 (ours)** | $50.5_{\pm 1.2}$ | $7.4_{\pm 1.2}$ | $15.4_{\pm 0.6}$ | $0.9_{\pm 0.3}$ |
> | GD + rank-2 | $57.5_{\pm 2.6}$ | $15.9_{\pm 1.8}$ | $18.1_{\pm 1.6}$ | $3.8_{\pm 1.4}$ |
> | GD + rank-3 | $60.4_{\pm 5.4}$ | $17.7_{\pm 5.0}$ | $16.8_{\pm 0.9}$ | $2.9_{\pm 0.6}$ |
> | GD + rank-4 | $55.7_{\pm 3.4}$ | $13.0_{\pm 2.0}$ | $17.0_{\pm 1.2}$ | $2.9_{\pm 1.2}$ |
> | GD + rank-5 | $55.6_{\pm 2.3}$ | $12.6_{\pm 1.6}$ | $17.1_{\pm 1.2}$ | $2.9_{\pm 1.0}$ |

---

> ### Author Response · Authors · 2025-11-18
>
> # "...Sensitivity to λ, μ-estimation schedule, and the SNR/timestep sampling strategy; (iii) replay buffer size; teacher choice (EMA vs last checkpoint) in distillation...."
>
> - $\lambda$: We evaluated different strength of EWC on CIFAR-10:
>
> | λ | FID |
> | --- | --- |
> | 20,000 | 58.47 |
> | **15,000** | **54.48** |
> | 10,000 | 55.98 |
> | 5,000 | 58.75 |
> - Other model- and training-specific hyperparameters:
> We used the default noise scheduler configuration from the original DDIM paper, together with Monte Carlo sampling over timesteps as in the DDIM training algorithm. For generative distillation, we followed the configuration from its original paper: we doubled the batch size to account for replay data (i.e., 128 current-task samples + 128 replay samples) and retained only the most recent teacher model. **We deliberately kept these configurations at their default settings in order to perform a controlled study of the effect of our rank-1 approximation, relying on hyperparameters that are widely used and empirically effective.**
>
>
> # "...No wall-clock/memory comparisons..."
>
> We refer to Table 3 in the appendix for the wall-clock comparison. For the low-rank baselines, training time roughly doubled due to the power iterations.
>
>
> # "...uses 32×32 ImageNet-1k;..."
>
> We emphasize that the primary goal of our ImageNet experiments is to examine whether our rank-1 approximation remains effective in a long-horizon continual learning setting. Specifically, we seek to test whether the rank-1 Fisher approximation is sufficiently accurate that errors from early tasks do not accumulate catastrophically over many subsequent tasks, while effectively mitigating distribution drift induced by replay.
>
> To make these long-horizon experiments computationally feasible, we adopt Downsampled ImageNet (Chrabaszcz et al., 2017) as a budget-friendly alternative to the full-resolution dataset. The ImageNet32x32 preserves key training characteristics of the original ImageNet
> while substantially reducing computational cost, which in turn enables faster experimental iteration and more extensive evaluation of our method.
>
> # "...I am wondering how the result would look like if λ=0 (no regularization)...."
>
> We argue that our baselines are well-tuned. When there is no regularization (i.e., plain continual learning), the performance is very poor:
> | Dataset | Final FID | Avg Forgetting |
> | --- | --- | --- |
> | CIFAR10 | 115.410 | 72.496 |
> | FMNIST | 103.955 | 88.009 |
> | IMAGENET32 | 265.902 | 67.864 |
> | MNIST | 102.642 | 93.700 |
>
> # "...Is this the first work about continual learning for diffusion model? If it is not, I expect the author to compare against other related work..."
>
> We argue that the majority of continual learning work with generative models has focused on generative replay techniques. For related work on continual learning with diffusion models specifically, we refer the reader to Section 2.3, paragraph 2. Generative distillation (GD) is a recent approach that improves upon standard generative replay for diffusion models in the image generation continual learning setting.
> In Table~1, GD denotes this Generative Distillation baseline.
> We refer to Zhang et al. (2024) for a more comprehensive benchmark of generative models on image-generation continual learning tasks; however, none of the benchmarked approaches is novel to diffusion models.
>
> - *Chrabaszcz, P., Loshchilov, I., & Hutter, F. (2017). A Downsampled Variant of ImageNet as an Alternative to the CIFAR datasets (Version 3). arXiv. https://doi.org/10.48550/ARXIV.1707.08819*
> - *Zhang, H., Zhou, J., Lin, H., Ye, H., Zhu, J., Wang, Z., Gao, L., Wang, Y., & Liang, Y. (2024). CLoG: Benchmarking Continual Learning of Image Generation Models (Version 1). arXiv. https://doi.org/10.48550/ARXIV.2406.04584*

---

### Meta-Review · Area_Chair_84Cd · 2026-01-08

**Summary:**

All reviewers suggested the rating of 6. They highlighted that it is well-written and offers a novel and insightful theoretical analysis regarding the gradient structure of diffusion models. They also liked the clean theory to practice bridge, convincing empirical validation, consistent and strong empirical improvements, near-zero forgetting on simple benchmarks and reduced forgetting on large-scale data, and its computationally efficiency.

**Reviewer Concerns:**

The main concerns are ablations missing/limited; no wall-clock/memory comparisons; heavy reliance on generative distillation; the most critical premise supporting this paper is assumption 1; the theorem considers the low SMR region; scope/scale limits: experiments are at 32×32 resolution only. Some of these are addressed in the rebuttal.

**Reviewer Scores:**

6,6,6,6.
I don't think the rebuttal would have reduced the scores so the reviewers' replies would not have changed the acceptance decision.

---

### Decision · Program_Chairs · 2026-01-26

Accept (Poster)